# SPECTRAL TRUNCATION KERNELS: NONCOMMUTATIVITY IN $C^*$-ALGEBRAIC KERNEL MACHINES

## ABSTRACT

$C^*$-algebra-valued kernels could pave the way for the next generation of kernel machines. To further our fundamental understanding of learning with $C^*$-algebraic kernels, we propose a new class of positive definite kernels based on the spectral truncation. We focus on kernels whose inputs and outputs are vectors or functions and generalize typical kernels by introducing the noncommutativity of the products appearing in the kernels. The noncommutativity induces interactions along the data function domain. We show that it is a governing factor leading to performance enhancement: we can balance the representation power and the model complexity. We also propose a deep learning perspective to increase the representation capacity of spectral truncation kernels. The flexibility of the proposed class of kernels allows us to go beyond previous separable and commutative kernels, addressing two of the foremost issues regarding learning in vector-valued RKHSs, namely the choice of the kernel and the computational cost.

## 1 INTRODUCTION

Kernel methods have been one of the most fundamental tools in machine learning (Schölkopf & Smola, 2001; Gretton et al., 2007; Hofmann et al., 2008; Muandet et al., 2017). They have been applied, for example, to ridge regression, principal component analysis, and support vector machine. Kernel methods are characterized by reproducing kernel Hilbert spaces (RKHSs), which are constructed by positive definite kernels. Typical positive definite kernels include the polynomial kernel, Gaussian kernel, and Laplacian kernel. Product kernels, which are constructed by the product of multiple kernels, have also been considered (Schölkopf & Smola, 2001; Thomas, 2008).

Standard positive definite kernels are scalar-valued, and are well-suited to learn scalar-valued functions. Kernel methods for vector- and function-valued outputs have also been investigated (Álvarez et al., 2012; Kadri et al., 2016). The kernels, in these cases, are instead operator-valued, and the associated feature space is vector-valued RKHSs (vvRKHSs) (Kadri et al., 2012; 2016; Minh et al., 2016). There are at least two challenges for vvRKHS methods: the computational cost and choice of kernels. A typical kernel is the separable kernel, which is defined by the product of a scalar-valued kernel and a positive semi-definite operator (Álvarez et al., 2012). Another typical kernel is the commutative kernel, which is defined only with the pointwise calculation of functions or vectors (Hashimoto et al., 2021). Although applying the separable and commutative kernels is computationally efficient, there is a crucial shortcoming for each kernel. Separable kernels identify dependencies between input and output variables separately, and cannot reflect information of input variables properly to output variables. The output is determined only by the global information of the input. On the other hand, commutative kernels only identify the pointwise (completely local) dependencies. Indeed, they are two extreme cases regarding the dependencies between input and output variables. Several attempts have been made to construct kernels that go beyond separable and commutative kernels. A typical nonseparable kernel is the transformable kernels, which is characterized by a map that can incorporate the information of input variables with the output variables. Huusari & Kadri (2021) proposed entangled kernels based on concepts from quantum computing, such as partial trace and entanglement. Hashimoto et al. (2023a) proposed to use the product of circulant matrices and general squared matrices to construct kernels. Using this kernel, one can generalize the convolution and capture the effect of interactions of different Fourier components. However, if we need an $m$-dimensional vector-valued outputs with these kernels, then we have to construct an $mN$ by $mN$ Gram matrix,

Table 1: Summary of the existing and the proposed kernels

| Type of kernels | Computational cost (ridge regression) | Extraction of *local* information | Extraction of *global* information |
|---|---|---|---|
| Separable | $O(mN^3)$ | × | ✓ |
| Commutative | $O(mN^3)$ | ✓ | × |
| Transformable | $O(m^3N^3)$ | ✓ | ✓ |
| **Proposed** (with RKHM) | $O(mn^2N^2 + mN^3)$ | ✓ | ✓ |

where $N$ is the sample size, and the computational cost is $O(m^3N^3)$ in general. Thus, with vvRKHSs, to go beyond separable and commutative kernels, the computational cost is significant.

In this work, we address the two challenges of the computational cost and the choice of kernels by introducing a new class of kernels based on the framework of reproducing kernel Hilbert $C^*$-module (RKHM) (Hashimoto et al., 2021). RKHM is a generalization of RKHS by means of $C^*$-algebra. $C^*$-algebra is a generalization of the space of complex values and has structures of the norm, product, and involution (Murphy, 1990; Lance, 1995). It unifies operators and functions. In this framework, kernels are generalized to $C^*$-algebra-valued kernel functions and allow us to consider function-valued kernels, leading to function-valued Gram matrices. By evaluating the values of the function-valued Gram matrix at $m$ different points, we obtain $m$ scalar-valued Gram matrices. This allows us to obtain an $m$-dimensional vector-valued outputs with the computational cost of $O(mN^3)$, which alleviates the dependency on $m$ from cubic to linear compared to the case of vvRKHSs with nonseparable kernels such as transformable kernels. We summarize the difference between the proposed and existing kernels in Table 1.

To obtain kernels going beyond the separable and commutative kernels with low computational cost, we propose a new class of $C^*$-algebra-valued positive definite kernels based on the spectral truncation, which has been discussed in the fields of noncommutative geometry and $C^*$-algebra (D'Andrea et al., 2014; van Suijlekom, 2021; Connes & van Suijlekom, 2021). The proposed kernels are parameterized by a natural number $n$ corresponding to the dimension of the truncated space. They can be applied to both vector and functional inputs. For vector inputs, we regard the elements of them as the values of functions. We approximate the input functions on the $n$-dimensional truncated space and obtain $n$ by $n$ Toeplitz matrices, whose $(i, j)$-entry depends only on $i - j$. Thus, $n$ describes the resolution of the discretization, and we call it the truncation parameter. Indeed, $n$ plays an important role from at least two perspectives. First, $n$ describes the noncommutativity of the kernel. Indeed, we show that the proposed kernels converge to the commutative kernels as $n$ goes to infinity. On the other hand, if $n = 1$, then the proposed kernels are separable kernel. Thus, we can control local and global dependencies through $n$. Second, the parameter $n$ controls the tradeoff between the representation power and the model complexity. We show that if $n$ is small, then the representation power is low, and the model complexity is small. On the other hand, if $n$ is large, then the representation power is high, and the model complexity is large. By introducing the parameter $n$ and setting $n$ to balance them, we obtain higher performance compared to the separable and commutative kernels. In the sense of these two perspective, in the setting of functional data (Jim Ramsay, 2005; Wang et al., 2016; Hashimoto et al., 2021), the proposed truncation kernels shed light on the good effects of *discretization* (setting $n$ as a finite number) on the learning process.

Our contributions are summarized as follows:
- We propose spectral truncation kernels, a new class of function-valued positive definite kernels that go beyond separable and commutative kernels with low computational cost. The proposed kernels are based on the spectral truncation and are indexed by a truncation parameter, which adjusts the global and local interactions along the data function domain by introducing the noncommutativity into the learning process. (see Definition 3.2 and Subsection 3.1)
- We derive a generalization bound for the learning problem in the RKHMs associated with the proposed kernels. The bound implies that the parameter $n$ controls the tradeoff between the representation power and the model complexity. (see Section 4)
- We propose a deep approach to gain the representation power. (see Section 6)

## 2 PRELIMINARIES

### 2.1 $C^*$-ALGEBRA AND REPRODUCING KERNEL HILBERT $C^*$-MODULE

$C^*$-algebra is a Banach space equipped with a product and an involution satisfying the $C^*$ identity (condition 3 below). It is a natural generalization of the space of complex numbers.

**Definition 2.1 ($C^*$-algebra)** *A set $\mathcal{A}$ is called a $C^*$-algebra if it satisfies the following conditions:*

*1. $\mathcal{A}$ is an algebra over $\mathbb{C}$ and equipped with a bijection $(\cdot)^* : \mathcal{A} \to \mathcal{A}$ that satisfies the following conditions for $\alpha, \beta \in \mathbb{C}$ and $a, b \in \mathcal{A}$:*

- $(\alpha a + \beta b)^* = \overline{\alpha} a^* + \overline{\beta} b^*$,    • $(ab)^* = b^* a^*$,    • $(a^*)^* = a$.

*2. $\mathcal{A}$ is a normed space endowed with a norm $\| \cdot \|_{\mathcal{A}}$, and for $a, b \in \mathcal{A}$, $\|ab\|_{\mathcal{A}} \leq \|a\|_{\mathcal{A}} \|b\|_{\mathcal{A}}$ holds. In addition, $\mathcal{A}$ is complete with respect to $\| \cdot \|_{\mathcal{A}}$.*

*3. For $a \in \mathcal{A}$, the $C^*$ identity $\|a^* a\|_{\mathcal{A}} = \|a\|_{\mathcal{A}}^2$ holds.*

*If there exists $a \in \mathcal{A}$ such that $ab = b = ba$ for any $b \in \mathcal{A}$, $a$ is called the unit and denoted by $1_{\mathcal{A}}$.*

In this paper, we focus on the $C^*$-algebra of continuous functions.

**Example 2.2** *Let $\mathbb{T} = \mathbb{R}/2\pi\mathbb{Z}$ be the torus and $C(\mathbb{T})$ be the space of continuous functions on $\mathbb{T}$. Then, $\mathcal{A} := C(\mathbb{T})$ is a $C^*$-algebra by means of the product: $(cd)(z) = c(z)d(z)$, involution: $c^*(z) = \overline{c(z)}$, and norm: $\|c\|_{\mathcal{A}} = \sup_{z \in \mathbb{T}} |c(z)|$ for $c, d \in \mathcal{A}$. The unit is the constant function $1_{\mathcal{A}} \equiv 1$.*

We now review basic notions regarding $C^*$-algebra. In the following, let $\mathcal{A}$ be a $C^*$-algbra.

**Definition 2.3 (Positive)** *An element $a$ of $\mathcal{A}$ is called* positive *if there exists $b \in \mathcal{A}$ such that $a = b^* b$ holds. For $a, b \in \mathcal{A}$, we write $a \leq_{\mathcal{A}} b$ if $b - a$ is positive.*

**Definition 2.4 (Infimum and minimum)** *For a subset $\mathcal{S}$ of $\mathcal{A}$, $a \in \mathcal{A}$ is said to be a* lower bound *with respect to the order $\leq_{\mathcal{A}}$, if $a \leq_{\mathcal{A}} b$ for any $b \in \mathcal{S}$. Then, a lower bound $c \in \mathcal{A}$ is said to be an* infimum *of $\mathcal{S}$, if $a \leq_{\mathcal{A}} c$ for any lower bound $a$ of $\mathcal{S}$. If $c \in \mathcal{S}$, then $c$ is said to be a* minimum *of $\mathcal{S}$.*

We now define RKHM. Let $\mathcal{X}$ be a non-empty set for data. To construct an RKHM, we first introduce $\mathcal{A}$-valued positive definite kernel.

**Definition 2.5 ($\mathcal{A}$-valued positive definite kernel)** *An $\mathcal{A}$-valued map $k : \mathcal{X} \times \mathcal{X} \to \mathcal{A}$ is called a positive definite kernel if it satisfies the following conditions:*

- $k(x, y) = k(y, x)^*$ *for $x, y \in \mathcal{X}$,*
- $\sum_{i,j=1}^{N} c_i^* k(x_i, x_j) c_j$ *is positive semi-definite for $n \in \mathbb{N}$, $c_i \in \mathcal{A}$, $x_i \in \mathcal{X}$.*

Let $\phi : \mathcal{X} \to \mathcal{A}^{\mathcal{X}}$ be the *feature map* associated with $k$, defined as $\phi(x) = k(\cdot, x)$ for $x \in \mathcal{X}$ and let $\mathcal{M}_{k,0} = \{\sum_{i=1}^{N} \phi(x_i) c_i | N \in \mathbb{N}, c_i \in \mathcal{A}, x_i \in \mathcal{X} \ (i = 1, \ldots, N)\}$. We can define an $\mathcal{A}$-valued map $\langle \cdot, \cdot \rangle_k : \mathcal{M}_{k,0} \times \mathcal{M}_{k,0} \to \mathcal{A}$ as

$$\left\langle \sum_{i=1}^{N} \phi(x_i) c_i, \sum_{j=1}^{M} \phi(y_j) d_j \right\rangle_k = \sum_{i=1}^{N} \sum_{j=1}^{M} c_i^* k(x_i, y_j) d_j,$$

which enjoys the reproducing property $\langle \phi(x), f \rangle_k = f(x)$ for $f \in \mathcal{M}_{k,0}$ and $x \in \mathcal{X}$.

The norm in $\mathcal{M}_{k,0}$ is defined as $\|f\|_k^2 = \| \langle f, f \rangle_k \|_{\mathcal{A}}$, and we can also define the $\mathcal{A}$-valued absolute value $|\cdot|_k$ as $|f|_k^2 = \langle f, f \rangle_k$. The *reproducing kernel Hilbert $\mathcal{A}$-module (RKHM) $\mathcal{M}_k$* associated with $k$ is defined as the completion of $\mathcal{M}_{k,0}$ with respect to $\| \cdot \|_k$. See, for example, the references (Lance, 1995; Murphy, 1990; Hashimoto et al., 2021) for more details about $C^*$-algebra and RKHM.

## 2.2 SPECTRAL TRUNCATION ON THE TORUS

The product in $C(\mathbb{T})$ is commutative. However, by approximating the multiplication of a function $x \in C(\mathbb{T})$ by a matrix, we can obtain a noncommutative product structure. Let $e_j$ be the Fourier function defined as $e_j(z) = e^{ijz}$ for $j \in \mathbb{Z}$ and $z \in \mathbb{T}$ and $M_x$ be the multiplication operator defined on $L^2(\mathbb{T})$ with respect to $x$. Here, $i$ is the imaginary unit, and $L^2(\mathbb{T})$ is the space of square-integrable functions on $\mathbb{T}$. Let $P_n$ be the orthogonal projection onto the $n$-dimensional subspace

Span$\{e_1, \ldots, e_n\}$. We approximate $M_x$ by $P_n M_x P_n$, i.e., by truncating the spectrum. Then, the $(j, l)$-entry of the representation matrix $R_n(x) \in \mathbb{C}^{n \times n}$ of $P_n M_x P_n$ is written as

$$R_n(x)_{j,l} = \langle e_j, M_x e_l \rangle_{L^2(\mathbb{T})} = \int_{\mathbb{T}} x(t) e^{-i(j-l)t} dt.$$

Since the $(j, l)$-entry of $R_n(x)$ depends only on $j - l$, $R_n(x)$ is a Toeplitz matrix (Gray, 2006). It is characterized only by a vector $r_n(x) \in \mathbb{C}^{2n-1}$, where the $(j - l)$th element of $r_n(x)$ is $R_n(x)_{j,l}$. Note that $R_n(x)_{j,l}$ is regarded as the $(j - l)$th Fourier component of $x$, and the vector $r_n(x)$ is the coordinate of the function $x$ projected on the space Span$\{e_{-(n-1)}, \ldots, e_{n-1}\}$.

For a matrix $A \in \mathbb{C}^{n \times n}$, let $S_n(A) \in C(\mathbb{T})$ be the function defined as $S_n(A)(z) = (1/n) \sum_{j,l=0}^{n-1} A_{j,l} e^{i(j-l)z}$, where $A_{j,l}$ is the $(j, l)$-entry of $A$. The map $S_n$ takes the representation matrix $R_n(x)$ back to a function that approximates the original function $x$. Indeed, we have

$$S_n(R_n(x))(z) = \int_{\mathbb{T}} x(t) \frac{1}{n} \sum_{j,l=0}^{n-1} e^{-i(j-l)t} e^{i(j-l)z} dt = \int_{\mathbb{T}} x(t) \frac{1}{n} \sum_{j=0}^{n-1} \sum_{l=-j}^{j} e^{il(z-t)} dt = x * F_n(z),$$

where $F_n(t) = (1/n) \sum_{j=0}^{n-1} \sum_{l=-j}^{j} e^{ilt}$ is the Fejér kernel and $*$ represents the convolution. If $n = 1$, then $F_n(z) = 1$, and as $n$ goes to infinity, $F_n$ goes to the delta function. More precisely, the following proposition holds (Brandolini & Travaglini, 1997; van Suijlekom, 2021), which implies that for each $z \in \mathbb{T}$, $S_n(R_n(x))(z)$ converges to $x(z)$ as $n \to \infty$.

**Proposition 2.6** *For each $z \in \mathbb{T}$, $x * F_n(z) \to x(z)$ as $n \to \infty$.*

A generalization of the Fejér kernel on $\mathbb{T}$ to that on $\mathbb{T}^q$ with respect to the sum over a polyhedron has been theoretically investigated (Travaglini, 1994; Brandolini & Travaglini, 1997). Let $P$ be a polyhedron and $jP = \{jz \mid z \in P\}$ for $j \in \mathbb{Z}$. For $t \in \mathbb{T}^q$, the Fejér kernel on $\mathbb{T}^q$ is defined as

$$F_n^{q,P}(t) = \frac{1}{n} \sum_{j=1}^{n-1} \sum_{r \in jP \bigcap \mathbb{Z}^q} e^{ir \cdot t}. \tag{1}$$

## 3 $C^*$-ALGEBRA-VALUED POSITIVE DEFINITE KERNEL WITH SPECTRAL TRUNCATION

In the following, we set the $C^*$-algebra $\mathcal{A}$ as $C(\mathbb{T})$. To obtain vector- or functional-valued outputs, applying vvRKHSs has been investigated. However, as we stated in Section 1, to go beyond the separable kernel and reach higher performance, the computational cost scales as $O(m^3 N^3)$ for $m$-dimensional outputs with $N$ samples. For more detailed explanation of the shortcomings of existing kernels with the framework of vvRKHSs, see Appendix A. To go beyond the separable and commutative kernels with lower computational cost, we consider function-valued ($\mathcal{A}$-valued) kernels and RKHMs. As we will discuss in the last part of Section 5, the computational cost depends on $mN^3$ if we use an $\mathcal{A}$-valued kernel. Hashimoto et al. (2021) proposed to use the $\mathcal{A}$-valued kernel $k$ where $k(\cdot, \cdot)(z)$ is a complex-valued positive definite kernel for all $z \in \mathbb{T}$. Combining it with typical existing kernels (Hashimoto et al., 2023a; Schölkopf & Smola, 2001; Kadri et al., 2012; Álvarez et al., 2012), we obtain the following three kinds of $\mathcal{A}$-valued kernels. Here, we consider the case where the set $\mathcal{X}$ for data is $\mathcal{X} \subseteq \mathcal{A}^d$.

**Example 3.1** *For $x = [x_1, \ldots, x_d], y = [y_1, \ldots, y_d] \in \mathcal{A}^d$ and $z \in \mathbb{T}$,*

1. *polynomial kernel $k^{\mathrm{poly},q}(x, y)(z) = \sum_{i=1}^{d} \alpha_i (\overline{x_i(z)} y_i(z))^q$,*

2. *product kernel $k^{\mathrm{prod},q}(x, y)(z) = \prod_{j=1}^{q} \overline{\tilde{k}_{1,j}(x(z), y(z))} \tilde{k}_{2,j}(x(z), y(z))$,*

3. *separable kernel $k^{\mathrm{sep},q}(x, y) = \tilde{k}(x, y) \prod_{j=1}^{q} (a_j^*)^q a_j^q$.*

*Here, $q \in \mathbb{N}$ is the degree of the products in the kernel, $\alpha_i \geq 0$ is the parameter of $k^{\mathrm{poly},q}$, and for $i = 1, 2$ and $j = 1, \ldots, q$, $\tilde{k}_{i,j} : \mathbb{C} \times \mathbb{C} \to \mathbb{C}$ and $\tilde{k} : \mathcal{A}^d \times \mathcal{A}^d \to \mathbb{C}$ are complex-valued continuous positive definite kernels. In addition, $a_j \in \mathcal{A}$ is the parameter of $k^{\mathrm{sep},q}$.*

We propose new $\mathcal{A}$-valued positive definite kernels by generalizing the typical kernels in Example 3.1. The kernels in Example 3.1 involve the product of functions in $\mathcal{A}$. For $x, y \in \mathcal{A}$, the product of $x$ and $y$ is commutative and is just the pointwise product defined as $(xy)(z) = x(z)y(z)$ (See Example 2.2). Thus, the kernels defined in Examples 3.1.1 and 3.1.2 do not induce the interactions along $z \in \mathbb{T}$, that is, $k(x, y)(z)$ is determined only by the values $x(z)$ and $y(z)$ at $z$. We call the kernel whose output at $z$ is determined only with $x(z)$ and $y(z)$ a commutative kernel. If the values $x(z_1)$ and $x(z_2)$, or $x(z_1)$ and $y(z_2)$ for different $z_1$ and $z_2$ are related, then the construction of the kernels only with this commutative product cannot extract that relationship. On the other hand, if we introduce the truncation parameter $n$ and transform $x$ and $y$ into Toeplitz matrices using the map $R_n$ defined in Subsection 2.2, then the product of $R_n(x)$ and $R_n(y)$ is noncommutative. Focusing on this feature, we define $\mathcal{A}$-valued kernels based on Example 3.1, but they have interactions along $z \in \mathbb{T}$.

**Definition 3.2** *With the notations in Example 3.1, for $x = [x_1, \ldots, x_d], y = [y_1, \ldots, y_d] \in \mathcal{A}^d$, let*

$$k_n^{\mathrm{poly},q}(x, y) = S_n \bigg( \sum_{i=1}^d \alpha_i (R_n(x_i)^*)^q R_n(y_i)^q \bigg),$$

$$k_n^{\mathrm{prod},q}(x, y) = S_n \bigg( \prod_{j=1}^q R_n(\tilde{k}_{1,j}(x, y))^* \prod_{j=1}^q R_n(\tilde{k}_{2,j}(x, y)) \bigg),$$

$$k_n^{\mathrm{sep},q}(x, y) = \tilde{k}(S_n(R_n(x)), S_n(R_n(y))) S_n \bigg( \prod_{j=1}^q R_n(a_j)^* \prod_{j=1}^q R_n(a_j) \bigg).$$

*Here, $A^*$ for a matrix $A$ is the adjoint, and we denote by $\tilde{k}_{i,j}(x, y)$ the map $z \mapsto \tilde{k}_{i,j}(x(z), y(z))$.*

**Remark 3.3** *Although the inputs of the kernels in Example 3.1 and Definition 3.2 are functions, we can also deal with vector inputs. For vector inputs, we can regard them as values of functions, and approximate the integral by the discrete sum.*

In Definition 3.2, to construct the kernels, we first project functions in $\mathcal{A}$ onto the space $\mathrm{Span}\{e_{-(n-1)}, \ldots, e_{n-1}\}$ using the map $R_n$ and obtain Toeplitz matrices. Then, we consider the product of the Toeplitz matrices. Note that the product of two Toeplitz matrices is not always a Toeplitz matrix. Finally, we apply $S_n$ to the matrix and take it back to the space $\mathrm{Span}\{e_{-(n-1)}, \ldots, e_{n-1}\}$ and obtain the output function of the kernel. Figure 3 in Appendix C schematically shows the construction of the kernel defined in Definition 3.2.

By the definitions of $S_n$ and $R_n$, we can show the following identity for $x_1, \ldots, x_q, y_1, \ldots, y_q \in \mathcal{A}$ (see the proof of Theorem 3.4 in Appendix E):

$$S_n \bigg( \prod_{j=1}^q R_n(x_j)^* \prod_{j=1}^q R_n(y) \bigg)(z) = \int_{\mathbb{T}^{2q}} \overline{x_1(t_1)} \cdots \overline{x_q(t_q)} y_1(t_{q+1}) \cdots y_q(t_{2q}) F_n^{2q,P}(z\mathbf{1} - t) \mathrm{d}t, \quad (2)$$

where $t = [t_1, \ldots, t_{2q}]$, $\mathbf{1} = [1, \ldots, 1]$, $P = \{[r_1, \ldots, r_{2q}] \in \mathbb{R}^{2q} \mid |\sum_{i=l}^m r_i| \leq 1, \ l \leq m\}$, and $F_n^{q,P}$ is the Fejér kernel on $\mathbb{T}^q$ defined in Eq. (1). Eq. (2) implies that the value of $S_n\big(\prod_{j=1}^q R_n(x_j)^* \prod_{j=1}^q R_n(y_j)\big)$ at $z$ is determined not only by $x_1(z), \ldots, x_q(z), y_1(z), \ldots, y_q(z)$, but also by $x_1(t_1), \ldots, x_q(t_q), y_1(t_{q+1}), \ldots, y_q(t_{2q})$ with different values $t_1, \ldots, t_{2q}$ from $z$. Note that $F_n^{q,P}(t)$ is a real-valued function since for $r \in P$, we have $-r \in P$. Thus, if a kernel in Example 3.1 is real-valued, then the corresponding kernel in Definition 3.2 is also real-valued.

### 3.1 CONVERGENCE AND INTERACTIONS

We first show that $k_n^{\mathrm{poly},q}$, $k_n^{\mathrm{prod},q}$, and $k_n^{\mathrm{sep},q}$ defined in Definition 3.2 converge to $k^{\mathrm{poly},q}$, $k^{\mathrm{prod},q}$, and $k^{\mathrm{sep},q}$ defined in Example 3.1 as $n$ goes to infinity, respectively. In this sense, the kernels $k_n^{\mathrm{poly},q}$, $k_n^{\mathrm{prod},q}$, and $k_n^{\mathrm{sep},q}$ are generalizations of typical kernels. In addition, note that if $n = 1$, then $R_n$ is just the averaging operation, and the output of the kernels are constant with respect to $z$. Thus, the proposed kernels are separable if $n = 1$ and are commutative if $n = \infty$. See Appendix B for more details. In this sense, the proposed kernels fill a gap between the separable and commutative kernels. In the following, all the proofs of theoretical statements are documented in Appendix E.

**Theorem 3.4** *For $x, y \in \mathcal{A}^d$ and $z \in \mathbb{T}$, $k_n^{\mathrm{poly},q}(x,y)(z) \to k^{\mathrm{poly},q}(x,y)(z)$, $k_n^{\mathrm{prod},q}(x,y)(z) \to k^{\mathrm{prod},q}(x,y)(z)$ as $n \to \infty$. If $x$ and $y$ are differentiable, then $k_n^{\mathrm{sep},q}(x,y)(z) \to k^{\mathrm{sep},q}(x,y)(z)$.*

As we can see in Eq. (2), if the kernel involves $q$ products of Toeplitz matrices generated by $R_n$, it is represented by a Fejér kernel on $\mathbb{T}^q$. To show Theorem 3.4, we apply the following lemmas.

**Lemma 3.5** *For $m \in \mathbb{N}$ and $j = 0, \ldots 2q$, let $Q_j^m = \{r' = [r'_1, \ldots, r'_{2q}] \in \mathbb{R}^{2q} \mid r'_i = r_i - r_{i-1} \ (i = 1, \ldots, 2q),\ 0 \le r_i \le m \ (i = 0, \ldots, 2q),\ r_j = m\}$ and $P = \{r = [r_1, \ldots, r_{2q}] \in \mathbb{R}^{2q} \mid |\sum_{i=l}^{k} r_i| \le 1,\ l \le k\}$. Then, we have $mP = \bigcup_{j=0}^{2q} Q_j^m$.*

**Lemma 3.6** *Let $P$ be a convex polyhedron. Let $F_n^{q,P}$ be the Fejér kernel on $\mathbb{T}^q$ defined as Eq. (1). Then, for any $g \in C(\mathbb{T}^q)$ and any $z \in \mathbb{T}^q$, $g * F_n^{q,P}(z) \to g(z)$ as $n \to \infty$.*

Since the product of Toeplitz matrices in $k_n^{\mathrm{poly},q}$, $k_n^{\mathrm{prod},q}$, and $k_n^{\mathrm{sep},q}$ are represented by the sum over the indices in $\bigcup_{j=0}^{2q} Q_j^m$, Lemma 3.5 explains why the Fejér kernel with the polyhedron $P = \{r \in \mathbb{R}^{2q} \mid |\sum_{i=l}^{k} r_i| \le 1,\ l \le k\}$ appears in Eq. (2). Lemma 3.6 generalizes the convergence of the Fejér kernel on $\mathbb{T}$ to that on $\mathbb{T}^q$, which is by Brandolini & Travaglini (1997). We set $g(z) = \overline{x_1(z_1) \cdots x_q(z_q)} y_1(z_{q+1}) \cdots y_q(z_{2q})$ for $z = [z_1, \ldots, z_{2q}]$, based on Eq. (2), and apply Lemmas 3.5 and 3.6 to show Theorem 3.4. See Appendix E for more details. Theorem 3.4 implies that the interactions along $z \in \mathbb{T}$ in the kernels become small as $n$ grows. This is because as $n$ goes to infinity, $F_n^{2q,P}$ goes to the delta function and by taking the convolution with the input function, it focuses more on local relationships between input and output functions as $n$ grows.

## 3.2 Positive definiteness

To construct an RKHM, which is introduced in Subsection 2.1, the kernel $k$ has to be positive definite. Thus, we investigate the positive definiteness of the proposed kernels. Regarding $k_n^{\mathrm{poly},q}(x,y)$, we can show the positive definiteness since $x_i$ and $y_i$ are separated by the product. Regarding $k_n^{\mathrm{sep}}(x,y)$, since $x$ and $y$ depend only on $\tilde{k}$, we can use the positive definiteness of $\tilde{k}$.

**Proposition 3.7** *The kernels $k_n^{\mathrm{poly},q}$ and $k_n^{\mathrm{sep},q}$ are positive definite.*

As for $k_n^{\mathrm{prod},q}(x,y)$, it depends on $x$ and $y$ through $\tilde{k}_{i,j}$, and we cannot separate $x$ and $y$ as products. Thus, we cannot show its positive definiteness. However, we can modify the kernel to become positive definite as follows.

**Proposition 3.8** *Let $\beta_n \ge -\min_{z \in \mathbb{T}^q} F_n^{2q,P}(z)$. Then, $\hat{k}_n^{\mathrm{prod},q}$ defined below is positive definite.*

$$\hat{k}_n^{\mathrm{prod},q}(x,y) = k_n^{\mathrm{prod},q}(x,y) + \beta_n \int_{\mathbb{T}^{2q}} \prod_{j=1}^{q} \overline{\tilde{k}_{1,j}(x(t_j), y(t_j))} \tilde{k}_{2,j}(x(t_{q+j}), y(t_{q+j})) \mathrm{d}t.$$

To set the value of the parameter $\beta_n$, we have the following bound.

**Lemma 3.9** *The Féjer kernel $F_n^{q,P}$ is bounded as $|F_n^{q,P}(z)| \le n^q$.*

**Remark 3.10** *We can set $\beta_n$ in $\hat{k}_n^{\mathrm{prod},q}$ as $n^q$ to guarantee the positive definiteness. However, even if we set a smaller $\beta_n$, $\hat{k}_n^{\mathrm{prod},q}$ may become positive definite. In addition, considering non-positive kernels has also been investigated (Ong et al., 2004; Canu et al., 2005). Indeed, in practical computations in Subsection 7, the algorithms can work even if we set $\beta_n < n^{2q}$. Deriving a tighter bound of $\beta_n$ or developing the theory for non-positive kernels for RKHMs is future work.*

## 4 Generalization Bound

We now focus on what impact the truncation parameter $n$ has on generalization. We provide a generalization bound allowing us to observe the tradeoff between the representation power and the model complexity associated with the proposed kernels. According to Lemma 4.2 in Mohri et al.

(2012), the generalization bound is described by the Rademacher complexity. We apply this lemma and derive a generalization bound.

Let $\Omega$ be a probability space with a probability measure $\mu$. Let $X_1, \ldots, X_N$ and $Y_1, \ldots, Y_N$ be samples from a distributions of $\mathcal{A}_0^d$-valued random variable $X$ and $\mathcal{A}_1$-valued random variable $Y$ on $\Omega$, respectively (i.e., for $z \in \mathbb{T}$, $X_i(z)$ is a sample from the distribution of $X(z)$). Here, $\mathcal{A}_0$ and $\mathcal{A}_1$ are subsets of $\mathcal{A}$. Let E be the Bochner integral on $\Omega$ with respect to $\mu$. Let $B > 0$ and $\mathcal{F} = \{f \in \mathcal{M}_k \mid \|f\|_k \le B\}$. Let $g : \mathbb{R} \times \mathbb{R} \to \mathbb{R}_+$ be an error function. Assume there exists $L > 0$ such that for $y \in \mathcal{A}_1$ and $z \in \mathbb{T}$, $x \mapsto g(x, y(z))$ is $L$-Lipschitz continuous. We derive the following generalization bound for the kernels defined in Section 3 based on Mohri et al. (2012); Hashimoto et al. (2023a) Here, to adapt to the generalization bound analysis, we assume the kernels are real-valued.

**Theorem 4.1** *Assume $k_n^{\mathrm{poly},q}$, $k_n^{\mathrm{prod},q}$, and $k_n^{\mathrm{sep},q}$ are real-valued. Let $D(k_n^{\mathrm{poly},q}, x) = \sum_{j=1}^d \alpha_j \|R_n(x_j)\|_{\mathrm{op}}^{2q}$, $D(\hat{k}_n^{\mathrm{prod},q}, x) = \prod_{j=1}^q (\|R_n(\tilde{k}_{1,j}(x,x))\|_{\mathrm{op}} \|R_n(\tilde{k}_{2,j}(x,x))\|_{\mathrm{op}}) + \beta_n C$, and $D(k_n^{\mathrm{sep},q}, x) = \tilde{k}(x,x) \prod_{j=1}^q \|R_n(a_j)\|_{\mathrm{op}}^2$ for $x \in \mathcal{A}_0^d$, where $\|\cdot\|_{\mathrm{op}}$ is the operator norm and $C = \prod_{j=1}^q \int_{\mathbb{T}} \tilde{k}_{1,j}(x(t), x(t)) \mathrm{d}t \int_{\mathbb{T}} \tilde{k}_{2,j}(x(t), x(t)) \mathrm{d}t$. Assume $\beta_n \le \beta_{n+1}$ for $\hat{k}_n^{\mathrm{prod},q}$. For $k_n = k_n^{\mathrm{poly},q}, \hat{k}_n^{\mathrm{prod},q}, k_n^{\mathrm{sep},q}$ and for any $\delta \in (0,1)$, with probability at least $1 - \delta$, we have*

$$\mathrm{E}[g(f(X), Y)] \le_{\mathcal{A}} \frac{1}{N} \sum_{i=1}^N g(f(X_i), Y_i) + 2L\frac{B}{N}\left(\sum_{i=1}^N D(k_n, X_i)\right)^{1/2} 1_{\mathcal{A}} + 3\sqrt{\frac{\log 1/\delta}{N}} 1_{\mathcal{A}} \quad (3)$$

*for any $f \in \mathcal{F}$. In addition, we have $D(k_n, x) \le D(k_{n+1}, x)$.*

The first term of the right-hand side of (3) is the empirical loss, and the second term represents the model complexity of the RKHM $\mathcal{M}_{k_n}$. There is a tradeoff between these terms. If $n$ is larger, then since the approximation space $\mathrm{Span}\{e_{-(n-1)}, \ldots, e_{n-1}\}$ in $\mathcal{A}$ is larger, the outputs of $k_n$ can represent more functions, which enables us to describe the dependency of outputs on inputs more. Indeed, if $n = 1$, then the output of $k_n$ is always a constant function, where the situation is the same as the existing separable kernel discussed in Section 1. Thus, the empirical loss (the first term) can become small if $n$ is large. On the other hand, according to Theorem 4.1, the complexity of $\mathcal{M}_{k_n}$ (the second term) is larger if $n$ is larger. An advantage of introducing the parameter $n$ to construct the new kernels based on the typical kernels in Example 3.1 is that we can control the empirical loss and the complexity through $n$. By setting a proper $n$, we can balance the two terms and make the expected loss $\mathrm{E}[g(f(X), Y)]$, the left-hand side of Eq. (3), small.

## 5 APPLICATION TO KERNEL RIDGE REGRESSION

We illustrate the effect of the proposed kernels by applying them to kernel ridge regression. Let $x_1, \ldots, x_N \in \mathcal{A}^d$ be input training samples and $y_1, \ldots, y_N \in \mathcal{A}$ be output training samples. We consider the case where we need the values of the output function evaluated at different $m$ points. We consider the RKHM $\mathcal{M}_k$ associated with $k = k_n^{\mathrm{poly},q}$, $k = k_n^{\mathrm{prod},q}$, or $k = k_n^{\mathrm{sep},q}$. Consider the typical minimization problem for regression, which is also considered by Hashimoto et al. (2023a):

$$\min_{f \in \mathcal{M}_k} \left( \sum_{i=1}^N |f(x_i) - y_i|_{\mathcal{A}}^2 + \lambda |f|_k^2 \right), \quad (4)$$

where $\lambda \ge 0$ is the regularization parameter. We apply the approximation version of the representer theorem for RKHMs over general $C^*$-algebras (see Appendix G). Then, the solution of the problem (4) is approximated by a vector having the form of $\sum_{i=1}^N \phi_i(x_i) c_i$. The problem is reduced to

$$\min_{c_j \in \mathcal{A}} \left( \sum_{i=1}^N \left| \sum_{j=1}^N k(x_i, x_j) c_j - y_i \right|_{\mathcal{A}}^2 + \lambda \left| \sum_{j=1}^N \phi(x_j) c_j \right|_k^2 \right)$$

$$= \min_{c_j \in \mathcal{A}} (\mathbf{c}^* \mathbf{G}^2 \mathbf{c} - \mathbf{c}^* \mathbf{G} \mathbf{y} - \mathbf{y}^* \mathbf{G} \mathbf{c} + \lambda \mathbf{c}^* \mathbf{G} \mathbf{c}), \quad (5)$$

where $\mathbf{G}$ is the $\mathcal{A}^{N \times N}$-valued Gram matrix whose $(i, j)$-entry is defined as $k(x_i, x_j) \in \mathcal{A}$, $\mathbf{c} = [c_1, \ldots, c_N]^T \in \mathcal{A}^N$, $\mathbf{y} = [y_1, \ldots, y_N]^T \in \mathcal{A}^N$. Then, the solution $\mathbf{c}$ of the problem (5) satisfies $\mathbf{y} = (\mathbf{G} + \lambda I)\mathbf{c}$, which means $\mathbf{y}(z) = (\mathbf{G}(z) + \lambda I)\mathbf{c}(z)$. Therefore, for each $z \in \mathbb{T}$, we obtain $\mathbf{c}(z)$ by computing $(\mathbf{G}(z) + \lambda I)^{-1}\mathbf{y}(z)$.

**Computational cost** The computational cost for the product of two $n$ by $n$ Toeplitz matrices is $O(n^2)$, and computing $S_n(A)(z)$ for an $n$ by $n$ matrix $A$ and $z \in \mathbb{T}$ costs $O(n^2)$. Thus, the cost for constructing $\mathbf{G}(z_1), \ldots, \mathbf{G}(z_m)$ for different $m$ points $z_1, \ldots, z_m \in \mathbb{T}$ with $k_n^{\mathrm{poly},q}$ or $\hat{k}_n^{\mathrm{prod},q}$ is $O((qn^2 + mn^2)N^2)$. As for the computation of $\mathbf{c}(z)$, we need to compute $(\mathbf{G}(z) + \lambda I)^{-1}\mathbf{y}(z)$, which costs $O(N^3)$. Thus, the total computational cost for obtaining $\mathbf{c}(z_1), \ldots, \mathbf{c}(z_m)$ is $O((q + m)n^2N^2 + mN^3)$. Note that the cost is linear with respect to $m$. This is by virtue of considering function-valued kernels and RKHMs. The situation is different if we use vvRKHSs. For vvRKHSs, to obtain $m$ values as an output, we typically apply a $\mathbb{C}^{O(m) \times O(m)}$-valued kernel. Then, the Gram matrix should be $mN$ by $mN$. Specifically, if the kernel is transformable, the computational cost is $O(m^3N^3)$. Thus, if $(q + m)n^2 < m^3N$, then the proposed kernels are more computationally efficient than the transformable kernels. See Appendices A and B for more details. For the proposed kernel, we can reduce the cost with respect to $N$, e.g., by applying Nyström method (Drineas & Mahoney, 2005) to each $\mathbf{G}(z_1), \ldots, \mathbf{G}(z_m)$. Regarding $n$ and $q$, one approach to reducing the cost with respect to them is investigating the method to approximate the Fejér kernel. However, the main contribution of this paper is to show the advantage of introducing the noncommutivity in kernels, and investigating computationally effective methods is future work.

## 6 DEEP MODEL WITH THE SPECTRAL TRUNCATION KERNELS

As we discussed in Section 4, there is a tradeoff between the representation power and the model complexity. In fact, there are two directions of the representation power: one is about the dependency of the outputs on the inputs discussed in Section 4, and the other is about the interactions along the data function domain. One shortcoming of the spectral truncation kernel is that to obtain a model with a high representation power in the sense of the dependency of the outputs on the inputs, the truncation parameter $n$ have to be large, which results in making the interactions along the data function domain weak. To gain the two types of representation powers at the same time, we consider the following deep model $f$ with $L$ spectral truncation kernels $k_n^1, \ldots, k_n^L$ and learnable parameters $c_i^1, \ldots, c_i^L \in \mathcal{A}$:

$$f(x) = \prod_{j=1}^{L} \left( \sum_{i=1}^{N} k_n^j(x, x_i) c_i^j \right). \tag{6}$$

We consider the case where $c_i^j$ in Eq. (6) is represented as a finite sum of functions in $\mathcal{A}$. In the following case, the representation power of this model grows exponentially.

**Proposition 6.1** *Let $k_n^j(x, x_i) = \sum_{l=-(n-1)}^{n-1} \tilde{d}_{i,l}^j e^{\mathrm{i}lz}$ with $\tilde{d}_{i,l}^j \in \mathbb{C}$. Assume $c_i^j$ is parameterized as $c_i^j(z) = \sum_{l=-(n-1)}^{n-1} d_{i,l}^j e_{\tau_j,l}$ with $d_{i,l}^j \in \mathbb{C}$ and $\tau_j \in \mathbb{R}$. Here, $e_{\tau_j,l} \in C(\mathbb{T})$ satisfies $e_{\tau_j,l}(z) = e^{\mathrm{i}\tau_j lz}$ for $z \in S$ for a subset $S$ of $\mathbb{T}$. Then, for $z \in S$, $f$ in Eq. (6) is represented as*

$$\sum_{l_1,\ldots,l_L=-(n-1)}^{n-1} \sum_{m_1,\ldots,m_L=-(n-1)}^{n-1} \sum_{i=1}^{N} \tilde{d}_{i,l_1}^1 \cdots \tilde{d}_{i,l_L}^L d_{i,l_1}^1 \cdots d_{i,l_L}^L e^{\mathrm{i}(l_1+\cdots+l_L+\tau_1 m_1+\cdots\tau_L m_L)z}.$$

*Let $V_{n,L} = \mathrm{Span}\{z \mapsto e^{\mathrm{i}(l_1+\cdots+l_L)z}e_{\tau_1,m_1}(z) \cdots e_{\tau_L,m_L}(z) \mid l_1,\ldots,l_L,m_1,\ldots,m_L \in \{-(n-1),\ldots,n-1\}\}$. If $\{\tau_1,\ldots,\tau_L\}$ is linearly independent over $\mathbb{Z}$, then $V_{n,L}$ is described by $2n + 1 + (2n+1)^L$ oscillating functions. In this sense, the representation power of the model $f$ grows exponentially with respect to the number of layers $L$.*

**Remark 6.2** *The exponential growth of the representation power is also observed for neural networks (Hanin & Rolnick, 2019). Deep learning with kernels has also been proposed (Laforgue et al., 2019; Hasimoto et al., 2023). Whereas the growth is obtained by the composition for these existing frameworks, it is obtained by the product for our case. Unlike the case of the existing frameworks, the representation power regarding $\mathcal{A}$ does not become high by the composition of functions.*

**Remark 6.3** *The function $f$ is in the RKHM associated with the kernel $\prod_{j=1}^{L} k^j$. Since the Rademacher complexity bound depends on the norm of the kernel, it can also grow exponentially with respect to $L$. Thus, for the deep setting, in the same manner as Theorem 4.1, we still have the tradeoff between the representation power and the model complexity. In this case, it is controlled by*

*both $n$ and $L$. The exponential growth of the Rademacher complexity bound (thus, the generalization bound) with respect to the number of layers is also observed for neural networks (Neyshabur et al., 2015; Bartlett et al., 2017; Golowich et al., 2018). Investigating how we can take advantage of the deep setting in the sense of the generalization is future work.*

# 7 NUMERICAL RESULTS

We show numerical results that illustrate the effects of the noncommutativity on the learning process.

## 7.1 EXPERIMENT WITH SYNTHETIC DATA

We first observed the behavior of the proposed kernels $k_n^{\text{poly},q}$, $\hat{k}_n^{\text{prod},q}$, and $k_n^{\text{sep},q}$ with different values of $n$ to observe the tradeoff between the representation power and model complexity discussed in Section 4 and compare them to the commutative kernels $k^{\text{poly},q}$, $k^{\text{prod},q}$, and $k^{\text{sep},q}$. We considered a regression task with synthetic data. For $i = 1, \ldots, N$, we generated input training samples as $x^i(z) = [\sin(0.01iz) + 0.01\xi_{z,i}, \cos(0.01iz) + 0.01\eta_{z,i}]$. We set the target function $f$ as $f(x)(z) = 3(\sin(\cos(\int_{z-\Delta}^{z+\Delta} x_1(t)\mathrm{d}t + \int_{z-\Delta}^{z+\Delta} x_2(t)\mathrm{d}t)))$, where $\Delta = 2\pi/30$, and generated training output samples $y^i(z) = f(x)(z) + 0.001\xi_{z,i}$. Here, $\xi_{z,i}$ and $\eta_{z,i}$ were sampled independently from the Gaussian distribution with mean 0 and standard deviation 1. We set the sample size $N$ as 1000. For the kernel $k_n^{\text{poly},q}$, we set $q = 1$ and $\alpha_1 = 1$. Note that $k^{\text{poly},q}$ is the linear kernel in this case. For the kernel $\hat{k}_n^{\text{prod},q}$, we set $q = 1$, $\tilde{k}_{1,1}(x,y) = \tilde{k}_{2,1}(x,y) = 2\pi e^{-|x-y|^2}$. We set $\beta_n = 1$ for $n < \infty$ and $\beta_\infty = 0$. For the kernel $k_n^{\text{sep},q}$, we set $\tilde{k}(x,y) = e^{-0.1 \cdot 2\pi^2(\|x_1-y_1\|_{L^2(\mathbb{T})}^2 + \|x_2-y_2\|_{L^2(\mathbb{T})}^2)}$, $q = 2$, and $a_1(z) = a_2(z) = 2\pi e^{\sin z}$. We estimated $f$ using kernel ridge regression. We applied the cross-validation grid search to find an optimal regularization parameter $\lambda$. Figure 1 (a-c) illustrates the test error. We can see that for $k_n^{\text{poly},q}$, the test error is the smallest when $n = 32$. It becomes large if $n$ is larger or smaller than 32. This is because when $n = 32$, the empirical error and the complexity in Eq. 3 are balanced, and the best possible expected error is obtained. We can see that even for the simplest kernel $k_n^{\text{poly},1}$, the proposed kernel ($n$ is finite) goes beyond the typical commutative kernel ($n = \infty$). We have similar results for the other two kernels. Note that $\hat{k}_\infty^{\text{prod},q}(x,y)(z) = \hat{k}^{\text{prod},q}(x,y)(z)$ only depends on $x(z)$ and $y(z)$. However, $f(x)(z)$ depends on $x(t)$ for $t \in [z-\Delta, z+\Delta]$. Thus, the test error becomes large when $n = \infty$. In addition, as discussed in Remark 3.10, $\hat{k}_n^{\text{prod},q}$ may not be positive definite. However, although we set $\beta_n$ as a small value, all the eigenvalues of the Gram matrix are positive in this case. See Appendix I.1 for more details.

We also compared the performance and the computational time with the proposed kernels to the existing operator-valued kernel with same setting as above to show the advantage of the proposed kernel in the sense of the computational efficiency and performance. For the vvRKHS, we used the nonseparable kernel $k(x,y) = \tilde{k}(x,y)[e^{-|x_i-y_j|^2}]_{i,j}$, where we replaced $S_n(\prod_{j=1}^q R_n(a_j)^* \prod_{j=1}^q R_n(a_j))$ with the nonseparable matrix $[e^{-|x_i-y_j|^2}]_{i,j}$. This kernel is the combination of the separable and transformable kernels and proposed by Lim et al. (2015). Table 2 shows the result. We can see that the proposed kernels outperform the existing typical nonseparable kernel, and the computational time with the proposed kernel is smaller than that with the existing kernel. Note that we have already confirmed the proposed kernels outperform the existing separable kernel $k_\infty^{\text{sep},q}$ in the above experiment (See Figure 1 (c)).

Finally, we observed the behavior of the deep model to show its high the representation power. We generated data in the same manner as above, but we set $N = 300$. We used the kernel $k_n^{\text{prod},q}$ and set $L = 1, 2, 3$, $\tau_1 = \sqrt{2}$, $\tau_2 = \sqrt{3}$, and $\tau_3 = \sqrt{5}/2$. We set $n$ so that the numbers of parameters are the same for all values of $L$. We used the loss function $\|\sum_{i=1}^N |f(x_i) - y_i|_k\|_{L^2(\mathbb{T})} + 0.1\||f|_k\|_{L^2(\mathbb{T})}$ for the deep model $f$ defined in Eq. (6). Figure 1 (d) shows the result. We can see as the number of layers increases, we can obtain a higher performance model even with the same number of parameters.

## 7.2 EXPERIMENT WITH MNIST

To check the performance of the proposed kernels for practical applications, we consider an image recovering task with MNIST (LeCun et al., 1998). For $i = 1, \ldots, N$, we generated input training

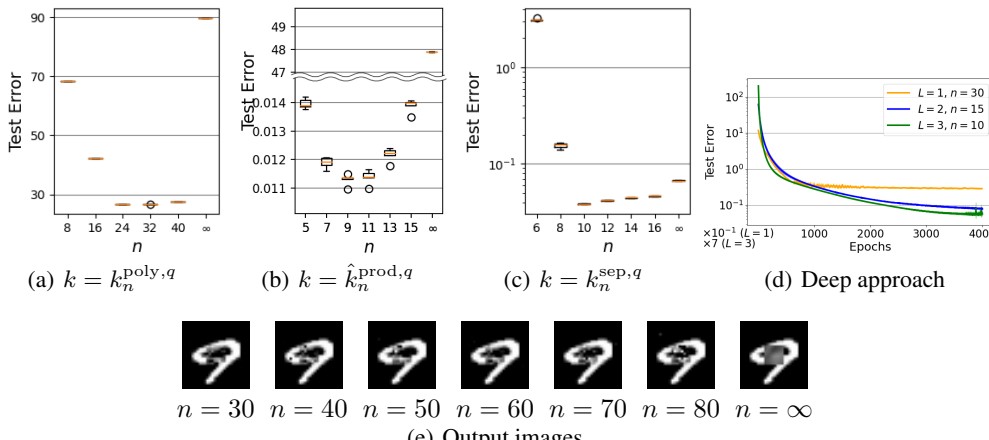

(a) $k = k_n^{\mathrm{poly},q}$      (b) $k = \hat{k}_n^{\mathrm{prod},q}$      (c) $k = k_n^{\mathrm{sep},q}$      (d) Deep approach

$n = 30$   $n = 40$   $n = 50$   $n = 60$   $n = 70$   $n = 80$   $n = \infty$

(e) Output images

Figure 1: (a-c) Test error of the regression task with different values of $n$. (Box plot of results of five independent runs with different values of noises $\xi_{z,i}$ and $\eta_{z,i}$ in $x^i(z)$ and $y^i(z)$.) (d) Test error of the regression task with deep approach with different $n$ and $L$. The parameters $n$ and $L$ are chosen so that the numbers of parameters are the same for all the cases. (e) Output images of the image recovering task with different values of $n$.

Table 2: Test error and CPU time of the regression task with RKHM and vvRKHS

| | Test Error | CPU Time (s) |
|---|---|---|
| RKHM, $k = k_n^{\mathrm{prod},q}$ ($n = 9$) | $0.0113 \pm 1.63 \times 10^{-5}$ | $149.3 \pm 2.392$ |
| RKHM, $k = k_n^{\mathrm{sep},q}$ ($n = 10$) | $0.0385 \pm 5.04388 \times 10^{-5}$ | $22.79 \pm 0.5742$ |
| vvRKHS (combination of transformable & separable kernels) | $0.0774 \pm 7.86 \ 10^{-5}$ | $570.4 \pm 14.87$ |

samples by setting all the $8 \times 8$ pixels in the middle of the images as 0. We set the output samples as the original images. We flatten the 2-dimensional $28 \times 28$ image to a vector in $\mathbb{R}^{784}$ and regard it as a discretized function on the space of pixels. We tried to estimate a function that transforms an image with the missing part into the original image. We applied kernel ridge regression with the regularization parameter $\lambda = 0.01$. We set the sample size $N$ as 200, and used the kernel $k_n^{\mathrm{prod},q}$ with $q = 1$ and $\tilde{k}_{1,1}(x,y) = \tilde{k}_{2,1}(x,y) = 2\pi e^{-0.1|x-y|^2}$. We set $\beta_n = 0.01$ for $n < \infty$ and $\beta_\infty = 0$. Figure 1 (e) shows the output images with different values of $n$. When $n = \infty$, i.e., the commutative kernel, we cannot recover the missing part since each pixel of the output is determined only with the corresponding pixel of the input, and we cannot obtain information of other pixels to estimate what is written in the image. When $n = 70$, we can recover the image the clearest.

## 8 CONCLUSION AND LIMITATION

In this paper, we proposed a new class of positive definite kernels for vector- or function-valued outputs based on the spectral truncation. The proposed kernels with the framework of RKHMs resolve two shortcomings of the framework of vvRKHSs for vector- or function-valued outputs at the same time: computational cost and choice of kernels. By considering function-valued positive definite kernels, not operator-valued ones, we can alleviate the computational cost. At the same time, we can introduce noncommutativity into the learning process and can induce interactions along the data function domain. We also showed that we can control the tradeoff between the representation power and the model complexity associated with the proposed kernels. In addition, we proposed a deep approach to obtain models with higher representation powers.

In the current setting, we focus on the $C^*$-algebra of continuous functions on the torus. Although this setting includes many important examples, e.g., periodic time-series data, continuous functions defined on a compact set in $\mathbb{R}$, finite-dimensional vectors, investigations for more general $C^*$-algebras allow us to apply the proposed kernel more general settings. As we discuss in Appendix H, we can generalize this setting and define positive definite kernels for more general inputs and outputs. In addition, as we discuss in Appendix D, there are many potential applications of the proposed kernel. However, theoretical and empirical investigations for these generalized settings and applications remain to be investigated in future work.

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

# APPENDIX

## A REVIEW OF EXISTING OPERATOR-VALUED KERNELS

We review existing operator-valued kernels and discuss their advantages and shortcomings in this section. We especially focus on the case $\mathcal{X} = \mathcal{A}^d$ and the application of kernel ridge regression in this paper. For more details about the operator-valued kernels, see, for example, (Álvarez et al., 2012).

Typical existing kernels are summirized as follows: Let $\tilde{k} : \tilde{\mathcal{X}} \times \tilde{\mathcal{X}} \to \mathbb{C}$ and $p : \mathbb{T} \times \mathbb{T} \to \mathbb{C}$ be complex-valued positive definite kernels and let $S : \mathbb{T} \times \mathcal{X} \to \tilde{\mathcal{X}}$. Consider a kernel $k : \mathcal{X} \times \mathcal{X} \to \mathcal{B}(L^2(\mathbb{T}))$ defined as

$$k(x, y)v(z) = \int_{\mathbb{T}} \tilde{k}(S(z, x), S(t, y))p(z, t)v(t)\mathrm{d}t. \tag{7}$$

In general, we have to construct a Gram matrix whose elements are operators. Typically, if we need an output function evaluated at $m$ points, then we discretize the operator on $O(m)$-dimensional space and obtain an $O(mN)$ by $O(mN)$ Gram matrix. Then, the computational cost for the kernel regression task for obtaining an output function evaluated at $m$ points is $O(N^3 m^3)$. Therefore, the computational cost is significant. As a special case of the kernel (7), the following two kernels are efficient in the sense of the computational cost.

**Separable kernel** If we set $\tilde{\mathcal{X}} = \mathcal{X}$ and $S(t, x) = x$, then the kernel $k$ defined in Eq. (7) is called a separable kernel. In other words, let $A \in \mathcal{B}(L^2(\mathbb{T}))$ be a Hermitian positive semi-definite operator defined as $Av(z) = \int_{\mathbb{T}} p(z, t)v(t)\mathrm{d}t$. The kernel $k : \mathcal{X} \times \mathcal{X} \to \mathcal{B}(L^2(\mathbb{T}))$ defined as

$$k(x, y)v(z) = \tilde{k}(x, y)Av(z)$$

is called a separable kernel.

For separable kernels, the computation is reduced to the computation of the Gram matrix of $\tilde{k}$. Thus, the computational cost for the kernel regression task for obtaining an output function evaluated at $m$ points is $O(N^3 m)$, where $N$ is the sample size. However, as we can see from the definition, it identifies dependencies between input and output variables separately and cannot reflect information of input variables to output variables. In the framework of vvRKHSs, the output function with an input $x$ is in the form of $\sum_{i=1}^{N} k(x, x_i)v_i$ for given samples $x_1, \ldots, x_N \in \mathcal{X}$, and some $v_1, \ldots, v_N \in L^2(\mathbb{T})$. Thus, with separable kernels, we cannot specify the relationship between the value of the output function at $z$ and the value $x(z)$ of the input $x$ at $z$ although they have strong connection in many cases. For example, if we have a time-series input $[x_1, \ldots, x_d] \in \mathcal{A}^d$ as explanatory variables and try to obtain an output function as a response variable, the values of $x_1(z), \ldots, x_d(z)$ at time $z$ is strongly related to the value of the output at time $z$. In this case, the separable kernels are not suitable for extracting the relationship between $x_1(z), \ldots, x_d(z)$ and the values of the output at $z$.

**Commutative kernel** If we set $\tilde{\mathcal{X}} = \mathbb{C}$, $S(t, x) = x(t)$, and $p(z, t) = 1$ $(z = t)$, $p(z, t) = 0$ $(z \neq t)$, then we have

$$k(x, y)v(z) = \tilde{k}(x(z), y(z))v(z).$$

We call this kernel the commutative kernel.

For commutative kernels, the computation is reduced to the computation at each $z$. Thus, the computational cost for the kernel regression task for obtaining an output function evaluated at $m$ points is $O(N^3 m)$. However, as we can see from the definition, it only identifies completely local relationship between the input function and the output function. The value of the output function at $z$ is determined only with the value of the input function at $z$. In the same example as separable kernels, the values of $x_1(z), \ldots, x_d(z)$ at time $z$ is strongly related to the value of the output at time $z$, but may also be related to $y(z + t)$ for $t \in [-T, T]$ for a small number $T$. In this case, the commutative kernels are not suitable for extracting the relationship between $x_1(z), \ldots, x_d(z)$ and the values of the output around $z$, not only exactly at $z$.

As we have seen, although the separable and commutative kernels are computationally efficient, they are two extreme cases regarding the description of the relationship between the input and output functions; the separable kernels identify dependencies between input and output variables separately (only with global information) and the commutative kernels only identify the commutative (completely local) dependencies. The following existing kernels fill a gap between the above two kernels, but computationally expensive.

**Transformable kernel** If we set $p(z, t) = 1$, then the kernel $k$ defined in Eq. (7) is called the transformable kernel. In this case, the kernels can identify the dependency between input and output variables through the map $S$. However, the computational cost for the kernel regression task for obtaining an output function evaluated at $m$ points is $O(N^3 m^3)$ as we discussed at the beginning of this section. In addition, we have to determine the map $S$, but it is not easy to interpret in general.

**Combination of separable and transformable kernels** Considering the sum of separable and transformable kernels and the product of these kernels have also been proposed. However, we have the same shortcomings of separable and transformable kernels.

## B CONNECTION BETWEEN THE PROPOSED KERNEL AND EXISTING KERNELS

The proposed function-valued kernels combined with the framework of RKHMs fill a gap between separable and commutative kernels with lower computational cost than transformable kernels. Indeed, we have the following observation:

**The case of $n = 1$** The proposed kernels $k_n^{\mathrm{poly},q}$, $k_n^{\mathrm{prod},q}$, and $k_n^{\mathrm{sep},q}$ are equivalent to separable kernels:

$$k_1^{\mathrm{poly},q}(x, y)(z) = \sum_{i=1}^{d} \alpha_i \left( \int_{\mathbb{T}} \overline{x_i(t)} \mathrm{d}t \right)^q \left( \int_{\mathbb{T}} y_i(t) \mathrm{d}t \right)^q,$$

$$k_1^{\mathrm{prod},q}(x, y)(z) = \prod_{j=1}^{q} \int_{\mathbb{T}} \overline{\tilde{k}_{1,j}(x(t), y(t))} \mathrm{d}t \prod_{j=1}^{q} \int_{\mathbb{T}} \tilde{k}_{2,j}(x(t), y(t)) \mathrm{d}t,$$

$$k_1^{\mathrm{sep},q}(x, y)(z) = \tilde{k} \left( \int_{\mathbb{T}} x(t) \mathrm{d}t, \int_{\mathbb{T}} y(t) \mathrm{d}t \right) \prod_{j=1}^{q} \int_{\mathbb{T}} \overline{a_j(t)} \mathrm{d}t \prod_{j=1}^{q} \int_{\mathbb{T}} a_j(t) \mathrm{d}t.$$

We can see that the output of the kernel does not depend on the variable $z$. The output function with an input $x$ of the kernel ridge regression is in the form of $\sum_{i=1}^{N} k(x, x_i)(z) c_i(z)$ for given samples $x_1, \ldots, x_N \in \mathcal{X}$ and some $c_1, \ldots, c_N \in C(\mathbb{T})$. Thus, the kernels identify dependencies between input and output functions separately. We can only capture global information of the input $x$ for obtaining the output.

**The case of $n = \infty$** The proposed kernels $k_n^{\mathrm{poly},q}$, $k_n^{\mathrm{prod},q}$ are equivalent to commutative kernels:

$$k_\infty^{\mathrm{poly},q}(x, y)(z) = \sum_{i=1}^{d} \alpha_i (\overline{x_i(z)} y_i(z))^q,$$

$$k_\infty^{\mathrm{prod},q}(x, y)(z) = \prod_{j=1}^{q} \overline{\tilde{k}_{1,j}(x(z), y(z))} \tilde{k}_{2,j}(x(z), y(z)).$$

The value of the output of the kernels at $z$ are determined only by $x(z)$ and $y(z)$. The kernels identify completely local dependencies between input and output functions.

**The case of $1 < n < \infty$** If $n$ is small, then we focus more on global information, and if $n$ is large, we focus more on local information. Indeed, as we discussed in Eq. (2), we have

$$k_n^{\mathrm{poly},q}(x, y)(z) = \int_{\mathbb{T}^{2q}} \overline{x_1(t_1)} \cdots \overline{x_q(t_q)} y_1(t_{q+1}) \cdots y_q(t_{2q}) F_n^{2q,P}(z\mathbf{1} - t) \mathrm{d}t,$$

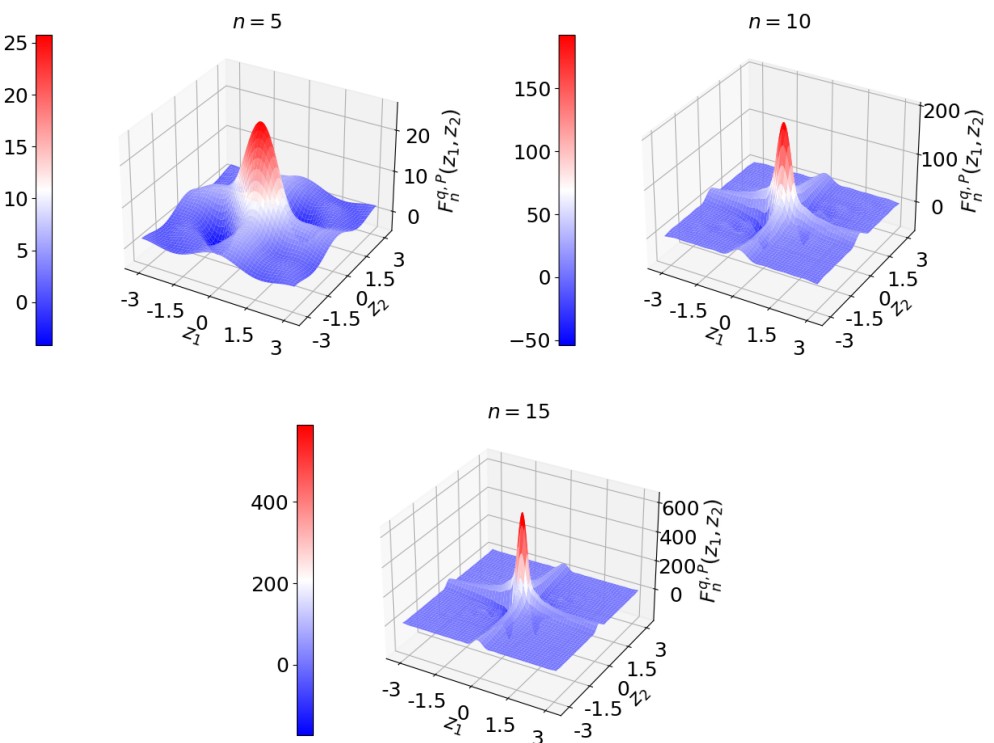

Figure 2: Féjer kernel $F_n^{2,P}$ for $n = 5, 10, 15$

$$k_n^{\text{prod},q}(x,y)(z) = \int_{\mathbb{T}^{2q}} \prod_{j=1}^{q} \overline{\tilde{k}_{1,j}(x(t_j), y(t_j))} \tilde{k}_{2,j}(x(t_{q+j}), y(t_{q+j})) F_n^{2q,P}(z\mathbf{1} - t)\mathrm{d}t,$$

where we assume $d = 1$ for simplify the notation. The proposed kernels are described by the convolution of the input function and the Féjer kernel $F_n^{2q,P}$. If $n = 1$, then $F_n^{2q,P}(z) = 1$, and as $n$ goes to infinity, $F_n^{2q,P}$ goes to the delta function. This means that if $n$ is small, then the convolution with $F_n^{2q,P}$ extract global information more than local information. On the other hand, if $n$ is large, then the convolution with $F_n^{2q,P}$ extract local information more than global information. We illustrate how $F_n^{2,P}$ changes along $n$ in Figure 2.

**Remark B.1** *We can determine optimal $n$ in the sense of the dependencies by observing the Féjer function $F_n^{2q,P}$. Since the proposed kernel is defined by the convolution of the input function with the Féjer kernel, the volume of the region where the value of the Féjer kernel is sufficiently large corresponds to the range of local dependencies. Thus, if we have a information of the local dependencies, then we can choose $n$ based on the values of $F_n^{2q,P}$.*

**Remark B.2** *The proposed kernels are composed of complex-valued kernels $\tilde{k}_{i,j}$ and $\tilde{k}$ in Definition 3.2. We can choose any kernel for $\tilde{k}_{i,j}$ and $\tilde{k}$, and the properties of the proposed kernels depend also on that choice. If we choose a kernel with parameters as $\tilde{k}_{i,j}$ or $\tilde{k}$ (such as a weighted sum of multiple kernels), then by optimizing the parameters, we can obtain a better kernel for given data or tasks. In addition, for function-valued kernels, in the same manner as the complex-valued kernels, the weighted sum of positive definite kernels and product of positive definite kernels are also a positive definite kernel. Thus, we can consider multiple proposed kernels and combine them with weight parameters. In that case, we can also optimize the parameters to obtain a better kernel for given data or tasks.*

**Computational cost**   As we discussed in the last paragraph of Section 5, the computational cost for the kernel regression task for obtaining an output function evaluated at $m$ points with the proposed

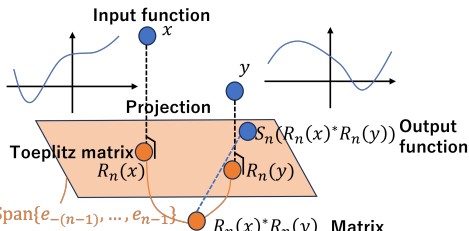

Figure 3: Overview of the construction of the simplest kernel $k_n^{\mathrm{poly},1}(x,y) = S_n(R_n(x)^*R_n(y))$

kernel $k_n^{\mathrm{poly},q}$ or $k_n^{\mathrm{prod},q}$ is $O((q+m)n^2N^2+mN^3)$. Thus, if $(q+m)n^2 < m^3N$, then the proposed kernels are more computationally efficient than nonseparable and noncommutative kernels, such as the transformable kernels.

## C  CONSTRUCTION OF THE PROPOSED KERNEL

In Definition 3.2, to construct the kernels, we first project functions in $\mathcal{A}$ onto the space $\mathrm{Span}\{e_{-(n-1)}, \ldots, e_{n-1}\}$ using the map $R_n$ and obtain Toeplitz matrices. Then, we consider the product of the Toeplitz matrices. Note that the product of two Toeplitz matrices is not always a Toeplitz matrix. Finally, we apply $S_n$ to the matrix and take it back to the space $\mathrm{Span}\{e_{-(n-1)}, \ldots, e_{n-1}\}$ and obtain the output function of the kernel.

Figure 3 schematically shows the construction of the simplest kernel $k_n^{\mathrm{poly},1}(x,y) = S_n(R_n(x)^*R_n(y))$ defined in Definition 3.2.

## D  APPLICATIONS: EXTRACTING LOCAL AND GLOBAL DEPENDENCIES OF FUNCTIONS

In Subsection 7.2, we considered applying the proposed kernel to an image recovering task. The application to image data is just an example, and there are more applications, which involves functions. We list two examples here.

**Time-series data analysis**  We can regard a time-series as a function on a time space. In many cases, a state at a certain time $z$ is influenced strongly by another state at the same time $z$, but also by the state around the time $z$. Since commutative kernels focus only on local information, we cannot describe these two states with commutative kernels. On the other hand, since separable kernels focus only on global information, we cannot describe the relationship of these two states at each time $z$. By applying the proposed kernels, we can extract global information, but also can focus on local information.

**Operator learning**  In the framework of operator learning, we obtain a solution of a partial differential equation as an output from an input function (such as initial condition or parameter of the equation). Thus, we construct a model where both of the input and output are functions. Applying kernel methods to operator learning has been proposed (Batlle et al., 2024). We can construct the model by solving a kernel ridge regression task. Another well-known operator learning method is neural operator. In the framework of neural operators, we apply integral operators to extract global information and apply local linear operators and local activation functions to extract local information. The proposed kernel enables us to do similar procedures for the operator learning with kernels. By considering the product of multiple proposed kernels with different values of $n$ or deep model with the proposed kernels with different values of $n$, we can extract both global and local information; we can extract global information using the kernel with small $n$ and extract local information using the one with large $n$ in the model.

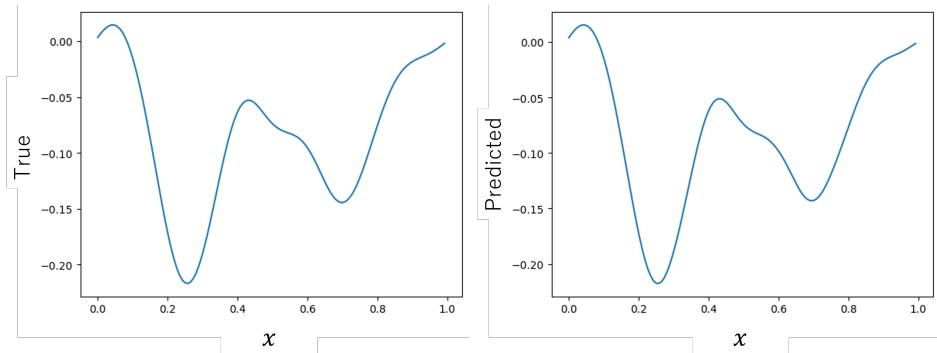

Figure 4: The true solution $w(\cdot, 1)$ and the predicted function $v$ of the Burgers' equation.

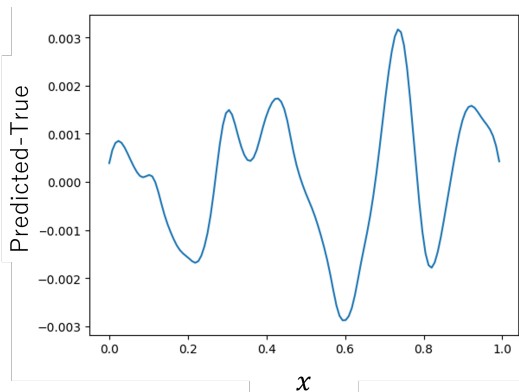

Figure 5: Pointwise error $v - w(\cdot, 1)$ of the true solution $w(\cdot, 1)$ and the predicted function $v$ of the Burgers' equation.

## D.1 NUMERICAL RESULTS

To show the availability of the proposed kernels to operator learning, we conducted an experiment. Consider the following Burgers' equation on $(0, 1] \times (0, 1]$.

$$\frac{\partial w}{\partial t} + w \frac{\partial w}{\partial x} = \nu \frac{\partial^2 w}{\partial x^2} \qquad x \in (0, 1) \times (0, 1],$$
$$w(x, 0) = u(x) \qquad x \in (0, 1),$$

where $\nu = 0.1$. We learned the operator mapping the initial condition $u$ to $w(\cdot, 1)$, the solution at time $t = 1$, by kernel ridge regression explained in Section 5. The training data and test data are generated by sampling the initial condition $u$ from the Gaussian distribution with mean 0 and the covariance $625(-\Delta + 25I)^{-2}$. We used equally spaced 128 points for computing the integral related to the input functions. The number of training samples is 1000 and that of test samples is 200. This problem is also considered by Batlle et al. (2024). We used the same $\hat{k}_n^{\mathrm{prod}, q}$ kernel as Subsection 7.1 with $n = 5$. We set the regularization parameter $\lambda$ as 0.01.

Figure 4 shows the true solution $w(\cdot, 1)$ and the predicted function $v$ by using the proposed kernel for the same test input function $u$ (initial condition). Figure 5 shows the error $v - w(\cdot, 1)$ of these two functions. The mean value of the pointwise error $1/128 \sum_{i=1}^{128} |v(x_i) - w(x_i, 1)|$ over all the test samples is $0.00157 \pm 7.079 \times 1e^{-6}$ (average $\pm$ standard deviation over five independent runs). We can see that the proposed kernel combined with kernel ridge regression properly predict the solution of the problem, and compared to the results in Subsection 4.3.1 by Batlle et al. (2024), the proposed kernel outperforms these existing results.

## E  PROOFS

We provide the proofs of theorems, propositions, and lemmas stated in the main text.

**Theorem 3.4**  *For $x, y \in \mathcal{A}^d$ and $z \in \mathbb{T}$, $k_n^{\mathrm{poly},q}(x,y)(z) \to k^{\mathrm{poly},q}(x,y)(z)$, $k_n^{\mathrm{prod},q}(x,y)(z) \to k^{\mathrm{prod},q}(x,y)(z)$ as $n \to \infty$. If $x$ and $y$ are differentiable, then $k_n^{\mathrm{sep},q}(x,y)(z) \to k^{\mathrm{sep},q}(x,y)(z)$.*

**Proof**  We show $k_n^{\mathrm{poly},q}(x,y)(z) \to k^{\mathrm{poly},q}(x,y)(z)$. The proofs for $k_n^{\mathrm{prod},q}(x,y)$ and $k_n^{\mathrm{sep},q}(x,y)$ are similar. We have

$$
k_n^{\mathrm{poly},q}(x,y)(z) = S_n\bigg( \sum_{i=1}^d \alpha_i (R_n(x_i)^*)^q R_n(y_i)^q \bigg)(z)
$$

$$
= \frac{1}{n} \sum_{i=1}^d \alpha_i \sum_{j,l=0}^{n-1} \sum_{r_1,\ldots,r_{2q}=0}^{n-1} R_n(x_i)^*_{j,r_1} \cdots R_n(x_i)^*_{r_{q-1},r_q} R_n(y_i)_{r_q,r_{q+1}} \cdots R_n(y_i)_{r_{2q-1},l} \mathrm{e}^{\mathrm{i}(j-l)z}
$$

$$
= \frac{1}{n} \sum_{i=1}^d \alpha_i \sum_{j,l=0}^{n-1} \sum_{r_1,\ldots,r_{2q-1}=0}^{n-1} \int_{\mathbb{T}} \cdots \int_{\mathbb{T}} \overline{x_i(t_1)} \cdots \overline{x_i(t_q)} y_i(t_{q+1}) \cdots y_i(t_{2q})
$$
$$
\cdot \mathrm{e}^{\mathrm{i}(r_1-j)t_1} \cdots \mathrm{e}^{\mathrm{i}(r_q-r_{q-1})t_q} \mathrm{e}^{\mathrm{i}(r_{q+1}-r_q)t_{q+1}} \cdots \mathrm{e}^{\mathrm{i}(l-r_{2q-1})t_{2q}} \mathrm{d}t_1 \cdots \mathrm{d}t_{2q} \mathrm{e}^{\mathrm{i}(j-l)z}
$$

$$
= \frac{1}{n} \sum_{i=1}^d \alpha_i \sum_{j,l=0}^{n-1} \sum_{r_1,\ldots,r_{2q-1}=0}^{n-1} \int_{\mathbb{T}} \cdots \int_{\mathbb{T}} \overline{x_i(t_1)} \cdots \overline{x_i(t_q)} y_i(t_{q+1}) \cdots y_i(t_{2q})
$$
$$
\cdot \mathrm{e}^{\mathrm{i}(r_1-j)(t_1-z)} \cdots \mathrm{e}^{\mathrm{i}(r_q-r_{q-1})(t_q-z)} \mathrm{e}^{\mathrm{i}(r_{q+1}-r_q)(t_{q+1}-z)} \cdots \mathrm{e}^{\mathrm{i}(l-r_{2q-1})(t_{2q}-z)} \mathrm{d}t_1 \cdots \mathrm{d}t_{2q}
$$

$$
= \frac{1}{n} \sum_{i=1}^d \alpha_i \sum_{m=0}^{n-1} \bigg( \sum_{j,l,r_1,\ldots,r_{2q-1}=0}^m - \sum_{j,l,r_1,\ldots,r_{2q-1}=0}^{m-1} \bigg) \int_{\mathbb{T}} \cdots \int_{\mathbb{T}} \overline{x_i(t_1)} \cdots \overline{x_i(t_q)} y_i(t_{q+1}) \cdots y_i(t_{2q})
$$
$$
\cdot \mathrm{e}^{\mathrm{i}(r_1-j)(t_1-z)} \cdots \mathrm{e}^{\mathrm{i}(r_q-r_{q-1})(t_q-z)} \mathrm{e}^{\mathrm{i}(r_{q+1}-r_q)(t_{q+1}-z)} \cdots \mathrm{e}^{\mathrm{i}(l-r_{2q-1})(t_{2q}-z)} \mathrm{d}t_1 \cdots \mathrm{d}t_{2q}
$$

$$
= \frac{1}{n} \sum_{i=1}^d \alpha_i \sum_{m=0}^{n-1} \sum_{\substack{j \vee l \vee r_1 \vee \ldots \vee r_{2q-1}=m \\ 0 \le j,l,r_1,\ldots,r_{2q-1} \le m}} \int_{\mathbb{T}} \cdots \int_{\mathbb{T}} \overline{x_i(t_1)} \cdots \overline{x_i(t_q)} y_i(t_{q+1}) \cdots y_i(t_{2q})
$$
$$
\cdot \mathrm{e}^{\mathrm{i}(r_1-j)(t_1-z)} \cdots \mathrm{e}^{\mathrm{i}(r_q-r_{q-1})(t_q-z)} \mathrm{e}^{\mathrm{i}(r_{q+1}-r_q)(t_{q+1}-z)} \cdots \mathrm{e}^{\mathrm{i}(l-r_{2q-1})(t_{2q}-z)} \mathrm{d}t_1 \cdots \mathrm{d}t_{2q}
$$

$$
= \frac{1}{n} \sum_{i=1}^d \alpha_i \sum_{j=0}^{n-1} \sum_{r \in jP \cap \mathbb{Z}^{2q}} \int_{\mathbb{T}} \cdots \int_{\mathbb{T}} \overline{x_i(t_1)} \cdots \overline{x_i(t_q)} y_i(t_{q+1}) \cdots y_i(t_{2q})
$$
$$
\cdot \mathrm{e}^{\mathrm{i}r_1(z-t_1)} \cdots \mathrm{e}^{\mathrm{i}r_q(z-t_q)} \mathrm{e}^{\mathrm{i}r_{q+1}(z-t_{q+1})} \cdots \mathrm{e}^{\mathrm{i}r_{2q}(z-t_{2q})} \mathrm{d}t_1 \cdots \mathrm{d}t_{2q}
$$

$$
= \sum_{i=1}^d \alpha_i g_i * F_n^{2q,P}(z),
$$

where the sum $\sum_{j,l,r_1,\ldots,r_{2q-1}=0}^{-1}$ is set as 0, $g_i(t) = \overline{x_i(t_1)} \cdots \overline{x_i(t_q)} y_i(t_{q+1}) \cdots y_i(t_{2q})$ for $t = [t_1,\ldots,t_{2q}] \in \mathbb{T}^{2q}$, and $P = \{ r = [r_1,\ldots,r_{2q}] \in \mathbb{R}^{2q} \mid |\sum_{i=l}^m r_i| \le 1, \, l \le m \}$. In addition, we set $j = r_0$ and $j = r_q$. The second to the last equality is derived by Lemma 3.5 below. Since $P$ is a convex polyhedron, $g_i * F_n^{2q,P}(z) \to g_i(z)$ as $n \to \infty$ by Lemma 3.6. For $k_n^{\mathrm{sep},q}$, we additionally use the fact $\lim_{n\to\infty} \tilde{k}(S_n(R_n(x)), S_n(R_n(y))) = \tilde{k}(\lim_{n\to\infty} S_n(R_n(x)), \lim_{n\to\infty} S_n(R_n(y)))$, which follows from the continuity of $\tilde{k}$, and the fact $S_n(R_n(x)) \to x$ uniformly as $n \to \infty$ if $x$ and $y$ are differentiable (van Suijlekom, 2021, Lemma 10). $\qquad\square$

**Lemma 3.5**  *For $m \in \mathbb{N}$ and $j = 0,\ldots 2q$, let $Q_j^m = \{ r' = [r_1',\ldots,r_{2q}'] \in \mathbb{R}^{2q} \mid r_i' = r_i - r_{i-1} \ (i = 1,\ldots,2q), \, 0 \le r_i \le m \ (i = 0,\ldots,2q), \, r_j = m \}$ and $P = \{ r = [r_1,\ldots,r_{2q}] \in \mathbb{R}^{2q} \mid |\sum_{i=l}^k r_i| \le 1, \, l \le k \}$. Then, we have $mP = \bigcup_{j=0}^{2q} Q_j^m$.*

**Proof** Let $r' \in Q_j^m$. For $i > j$, we have $r_i = r_i' + r_{i-1} = r_i' + (r_{i-1}' + r_{i-2}) = \cdots = r_i' + \cdots + r_{j+1}' + m$, and for $i < j$, we have $-r_i = r_{i+1}' - r_{i+1} = r_{i+1}' + (r_{i+2}' - r_{i+2}) = \cdots = r_{i+1}' + \cdots + r_j' - m$. Therefore, we have

$$
\bigcup_{j=0}^{2q} Q_j^m
$$

$$
= \bigcup_{j=0}^{2q} \left\{ r \in \mathbb{R}^{2q} \,\middle|\, r_i = \sum_{l=j+1}^{i} r_l' + m \ (j < i \le 2q), \ r_i = - \sum_{l=i+1}^{j} r_l' + m \ (0 \le i < j), \right.
$$
$$
\left. 0 \le r_i \le m \ (i = 0, \ldots, j-1, j+1, \ldots 2q) \right\}
$$

$$
= \bigcup_{j=0}^{2q} \left\{ r' \in \mathbb{R}^{2q} \,\middle|\, -m \le \sum_{l=j+1}^{i} r_l' \le 0 \ (j < i \le 2q), \ 0 \le \sum_{l=i+1}^{j} r_l' \le m \ (0 \le i < j) \right\}
$$

$$
= \left\{ r \in \mathbb{R}^{2q} \,\middle|\, \left| \sum_{i=l}^{k} r_i \right| \le 1, \ l \le k \right\},
$$

which completes the proof of the lemma. $\qquad\square$

**Proposition 3.7** *The kernels $k_n^{\mathrm{poly},q}$ and $k_n^{\mathrm{sep},q}$ are positive definite.*

**Proof** Let $x_1, \ldots, x_N \in \mathcal{A}^d$ and $d_1, \ldots, d_N \in \mathcal{A}$. Then, we have

$$
\left( \sum_{j,l=1}^{N} d_j^* k_n^{\mathrm{poly},q}(x_j, x_l) d_l \right)(z)
$$

$$
= \sum_{j,l=1}^{N} \overline{d_j(z)} \sum_{j',l'=1}^{n} \sum_{i=1}^{d} \alpha_i \sum_{m=1}^{n} \left( (R_n(x_{i,j})^*)^q \right)_{j',m} \left( R_n(x_{i,l})^q \right)_{m,l'} \mathrm{e}^{\mathrm{i}(j'-l')z} d_l(z)
$$

$$
= \sum_{i=1}^{d} \alpha_i \sum_{m=1}^{n} \left| \sum_{j=1}^{N} \sum_{j'=1}^{n} d_j(z) \left( (R_n(x_{i,j}))^q \right)_{m,j'} \mathrm{e}^{-\mathrm{i}j'z} \right|^2 \ge 0
$$

for $z \in \mathbb{T}$. In addition, let $\tilde{x}_{n,j} = S_n(R_n(x_j))$. Then, we have

$$
\left( \sum_{j,l=1}^{N} d_j^* k_n^{\mathrm{sep},q}(x_j, x_l) d_l \right)(z)
$$

$$
= \sum_{j,l=1}^{N} \overline{d_j(z)} \tilde{k}(\tilde{x}_{n,j}, \tilde{x}_{n,l}) d_l(z) \sum_{j',l'=1}^{n} \sum_{m_1,\ldots,m_{2q-1}=1}^{n}
$$

$$
\left( (R_n(a)^*) \right)_{j',m_1} \left( (R_n(a)^*) \right)_{m_1,m_2} \cdots \left( (R_n(a)^*) \right)_{m_{q-1},m_q} \left( R_n(a) \right)_{m_q,m_{q+1}} \cdots \left( R_n(a) \right)_{m_{2q-1},l'} \mathrm{e}^{\mathrm{i}(j'-l')z}
$$

$$
= \sum_{j,l=1}^{N} \overline{d_j(z)} \tilde{k}(\tilde{x}_{n,j}, \tilde{x}_{n,l}) d_l(z)
$$

$$
\sum_{m_q=1}^{n} \left| \sum_{j'=1}^{n} \sum_{m_1,\ldots,m_{q-1}=1}^{n} \left( R_n(a) \right)_{m_q,m_1} \cdots \left( R_n(a) \right)_{m_{q-2},m_{q-1}} \left( R_n(a) \right)_{m_{q-1},j'} \mathrm{e}^{-\mathrm{i}j'z} \right|^2
$$

$$
\ge 0.
$$

Thus, $k_n^{\mathrm{poly},q}$ and $k_n^{\mathrm{sep},q}$ are positive definite. $\qquad\square$

**Proposition 3.8** *Let $\beta_n \geq -\min_{z \in \mathbb{T}^q} F_n^{2q,P}(z)$. Then, $\hat{k}_n^{\mathrm{prod},q}$ defined below is positive definite.*

$$\hat{k}_n^{\mathrm{prod},q}(x,y) = k_n^{\mathrm{prod},q}(x,y) + \beta_n \int_{\mathbb{T}^{2q}} \prod_{j=1}^{q} \overline{\tilde{k}_{1,j}(x(t_j),y(t_j))}\tilde{k}_{2,j}(x(t_{q+j}),y(t_{q+j}))\mathrm{d}t.$$

**Proof** Let $x_1, \ldots, x_N \in \mathcal{A}^d$ and $d_1, \ldots, d_N \in \mathcal{A}$. Then, we have

$$\left( \sum_{j,l=1}^{N} d_j^* \hat{k}_n^{\mathrm{prod},q}(x_j, x_l) d_l \right)(z)$$

$$= \sum_{m,l=1}^{N} \int_{\mathbb{T}^{2q}} \overline{d_m(z)} \left( \prod_{j=1}^{q} \overline{\tilde{k}_{1,j}(x_m(t_j), x_l(t_j))}\tilde{k}_{2,j}(x_m(t_{q+j}), x_l(t_{q+j})) \right) d_l(z) F_n^{2q,P}(z\mathbf{1} - t)\mathrm{d}t$$

$$+ \beta_n \sum_{m,l=1}^{N} \int_{\mathbb{T}^{2q}} \overline{d_m(z)} \left( \prod_{j=1}^{q} \overline{\tilde{k}_{1,j}(x_m(t_j), x_l(t_j))}\tilde{k}_{2,j}(x_m(t_{q+j}), x_l(t_{q+j})) \right) d_l(z)\mathrm{d}t$$

$$= \sum_{m,l=1}^{N} \int_{\mathbb{T}^{2q}} \overline{d_m(z)} \left( \prod_{j=1}^{q} \overline{\tilde{k}_{1,j}(x_m(t_j), x_l(t_j))}\tilde{k}_{2,j}(x_m(t_{q+j}), x_l(t_{q+j})) \right) d_l(z)(F_n^{2q,P}(z\mathbf{1} - t) + \beta_n)\mathrm{d}t$$

$$\geq 0$$

for $z \in \mathbb{T}$. The last inequality is derived since the map $([x_1, \ldots, x_{2q}], [y_1, \ldots, y_{2q}]) \mapsto \prod_{j=1}^{q} \overline{\tilde{k}_{1,j}(x_{2j-1}, y_{2j-1})}\tilde{k}_{2,j}(x_{2j}, y_{2j})$ is positive definite and $F_n^{2q,P}(z\mathbf{1} - t) + \beta_n \geq 0$. $\square$

**Lemma 3.9** *We have $|F_n^{q,P}(z)| \leq n^q$.*

**Proof** The number of terms in $F_n^{q,P}$ is equal to the number of terms in $\sum_{m,l=0}^{n-1}(T_1 \cdots T_{2q})_{m,l}$, where $T_1, \ldots, T_{2q}$ are Toeplitz matrices. Thus, we have

$$|F_n^{q,P}(z)| \leq \frac{1}{n}n^{q+1} = n^q.$$

$\square$

Let $\Omega$ be a probability space with a probability measure $\mu$. Let $X_1, \ldots, X_N$ and $Y_1, \ldots, Y_N$ be samples from a distributions of $\mathcal{A}_0^d$-valued random variable $X$ and $\mathcal{A}_1$-valued random variable $Y$ on $\Omega$, respectively (i.e., for $z \in \mathbb{T}$, $X_i(z)$ is a sample from the distribution of $X(z)$). Here, $\mathcal{A}_0$ and $\mathcal{A}_1$ are subsets of $\mathcal{A}$.

**Theorem 4.1** *Assume $k_n^{\mathrm{poly},q}$, $k_n^{\mathrm{prod},q}$, and $k_n^{\mathrm{sep}\,q}$ are real-valued. Let $D(k_n^{\mathrm{poly},q}, x) = \sum_{j=1}^{d} \alpha_j \|R_n(x_j)\|_{\mathrm{op}}^{2q}$, $D(\hat{k}_n^{\mathrm{prod},q}, x) = \prod_{j=1}^{q}(\|R_n(\tilde{k}_{1,j}(x,x))\|_{\mathrm{op}}\|R_n(\tilde{k}_{2,j}(x,x))\|_{\mathrm{op}}) + \beta_n C$, and $D(k_n^{\mathrm{sep},q}, x) = \tilde{k}(x,x)\prod_{j=1}^{q} \|R_n(a_j)\|_{\mathrm{op}}^2$ for $x \in \mathcal{A}_0^d$, where $\|\cdot\|_{\mathrm{op}}$ is the operator norm and $C = \prod_{j=1}^{q} \int_{\mathbb{T}} \tilde{k}_{1,j}(x(t), x(t))\mathrm{d}t \int_{\mathbb{T}} \tilde{k}_{2,j}(x(t), x(t))\mathrm{d}t$. Assume $\beta_n \leq \beta_{n+1}$ for $\hat{k}_n^{\mathrm{prod},q}$. For $k_n = k_n^{\mathrm{poly},q}, \hat{k}_n^{\mathrm{prod},q}, k_n^{\mathrm{sep},q}$ and for any $\delta \in (0,1)$, with probability at least $1 - \delta$, we have*

$$\mathrm{E}[g(f(X),Y)] \leq_{\mathcal{A}} \frac{1}{N}\sum_{i=1}^{N} g(f(X_i), Y_i) + 2L\frac{B}{N}\left( \sum_{i=1}^{N} D(k_n, X_i) \right)^{1/2} 1_{\mathcal{A}} + 3\sqrt{\frac{\log 1/\delta}{N}}1_{\mathcal{A}}$$

*for any $f \in \mathcal{F}$. In addition, we have $D(k_n, x) \leq D(k_{n+1}, x)$.*

To show Theorem 4.1, we use the $\mathcal{A}$-valued Rademacher complexity defined by Hashimoto et al. (2023a). Let $\sigma_1, \ldots, \sigma_N$ be i.i.d. Rademacher variables, which take their values on $\{-1, 1\}$. For a

real-valued function class $\mathcal{F}$ and $\mathbf{x} = [x_1, \ldots, x_N] \in (\mathcal{A}^d)^N$, the empirical Rademacher complexity $\hat{R}_N(\mathbf{x}, \mathcal{F})$ is defined by $\hat{R}_N(\mathbf{x}, \mathcal{F}) = \mathrm{E}[\sup_{f \in \mathcal{F}} \sum_{i=1}^N f(x_i)\sigma_i]/N$. Here, $\mathrm{E}$ is the integration on $\Omega$ with respect to $\mu$. Similar to the case of the standard Rademacher complexity, an $\mathcal{A}$-valued version of the Rademacher complexity is defined as $\hat{R}_{\mathcal{A},N}(\mathbf{x}, \mathcal{F}) = \mathrm{E}[\sup_{f \in \mathcal{F}}^{\mathcal{A}} \sum_{i=1}^N |f(x_i)^*\sigma_i|_{\mathcal{A}}]/N$ for an $\mathcal{A}$-valued function class $\mathcal{F}$. Here, $|a|_{\mathcal{A}} = (a^*a)^{1/2}$ for $a \in \mathcal{A}$, $\sup^{\mathcal{A}}$ is the supremum in the sense of the order in $\mathcal{A}$ (see Definition 2.4), and the integral $\mathrm{E}$ means the Bochner integral in this case.

Indeed, in our case, the $\mathcal{A}$-valued Rademacher complexity is represented by the standard Rademacher complexity, and the argument is reduced to evaluate the standard Rademacher complexity. In the following, we denote by $\mathcal{A}_+$ the subset of $\mathcal{A}$ composed of all positive elements in $\mathcal{A}$.

**Proposition E.1** *Let $\mathcal{F}$ be an $\mathcal{A}$-valued function class. Assume for any $x \in \mathcal{A}_0^d$ and $z \in \mathbb{T}$, $f(x)(z) \in \mathbb{R}$. Then, we have $\hat{R}_N(\mathbf{x}, \mathcal{F}(z)) = \hat{R}_{\mathcal{A},N}(\mathbf{x}, \mathcal{F})(z)$, where $\mathcal{F}(z) = \{x \mapsto (f(x))(z) \mid f \in \mathcal{F}\}$.*

**Proof** We first show $(\sup_{a \in \mathcal{S}}^{\mathcal{A}} a)(z) = \sup_{a \in \mathcal{S}} a(z)$ for $\mathcal{S} \subseteq \mathcal{A}_+$ for any $z \in \mathbb{T}$. Let $a \in \mathcal{S}$. Since $a(z) \le \sup_{a \in \mathcal{S}} a(z)$ for any $z \in \mathbb{T}$, we have $a \le_{\mathcal{A}} b$, where $b \in \mathcal{A}$ is defined as $b(z) = \sup_{a \in \mathcal{S}} a(z)$. Thus, $b$ is an upper bound of $\mathcal{S}$, and we have $\sup_{a \in \mathcal{S}}^{\mathcal{A}} a \le_{\mathcal{A}} b$, which means $(\sup_{a \in \mathcal{S}}^{\mathcal{A}} a)(z) \le \sup_{a \in \mathcal{S}} a(z)$ for any $z \in \mathbb{T}$. Conversely, since $a \le_{\mathcal{A}} \sup_{a \in \mathcal{S}}^{\mathcal{A}} a$ for $a \in \mathcal{S}$, we have $a(z) \le (\sup_{a \in \mathcal{S}}^{\mathcal{A}} a)(z)$ for any $z \in \mathbb{T}$. Thus, we have $\sup_{a \in \mathcal{S}} a(z) \le (\sup_{a \in \mathcal{S}}^{\mathcal{A}} a)(z)$.

Therefore, we have

$$\hat{R}_{\mathcal{A},N}(\mathbf{x}, \mathcal{F})(z) = \frac{1}{N} \mathrm{E}\left[\sup_{f \in \mathcal{F}}^{\mathcal{A}} \left| \sum_{i=1}^N f(x_i)^*\sigma_i \right|_{\mathcal{A}} \right](z) = \frac{1}{N} \mathrm{E}\left[ \left(\sup_{f \in \mathcal{F}}^{\mathcal{A}} \left| \sum_{i=1}^N f(x_i)^*\sigma_i \right|_{\mathcal{A}} \right)(z) \right]$$

$$= \frac{1}{N} \mathrm{E}\left[ \sup_{f \in \mathcal{F}(z)} \left| \sum_{i=1}^N \overline{f(x_i)}\sigma_i \right| \right] = \frac{1}{N} \mathrm{E}\left[ \sup_{f \in \mathcal{F}(z)} \left| \sum_{i=1}^N f(x_i)\sigma_i \right| \right] = \hat{R}_N(\mathbf{x}, \mathcal{F}(z)),$$

where the third equality is given by the identity $|a|_{\mathcal{A}}(z) = |a(z)|$, and the forth equality is satisfied since $\sigma_i$ is real-valued. $\qquad\qquad\square$

We obtain the following lemma directly by Lemma 4.2 in Mohri et al. (2012).

**Lemma E.2** *Let $\mathcal{F}$ be an $\mathcal{A}$-valued function class. Let $g : \mathbb{R} \times \mathbb{R} \to \mathbb{R}_+$ be an error function. Assume there exists $L > 0$ such that for $y \in \mathcal{A}_1$ and $z \in \mathbb{T}$, $x \mapsto g(x, y(z))$ is $L$-Lipschitz continuous. Assume also for any $x \in \mathcal{A}_0^d$ and $z \in \mathbb{T}$, $f(x)(z) \in \mathbb{R}$. Then, we have*

$$\frac{1}{N} \mathrm{E}\left[ \sup_{f \in \mathcal{F}} \sum_{i=1}^N g(f(x_i)(z), y_i(z))\sigma_i \right] = \frac{1}{N} \mathrm{E}\left[ \sup_{f \in \mathcal{F}(z)} \sum_{i=1}^N g(f(x_i), y_i(z))\sigma_i \right] \le L\hat{R}_N(\mathbf{x}, \mathcal{F}(z)).$$

We apply the following lemma to obtain the inequality (3).

**Lemma E.3** *Let $\mathcal{F}$ be an $\mathcal{A}$-valued function class. Let $g$ be the same map defined in Lemma E.2. For any $f \in \mathcal{F}$ and any $\delta \in (0, 1)$, with probability at least $1 - \delta$, we have*

$$\mathrm{E}[g(f(X), Y)](z) \le \frac{1}{N} \sum_{i=1}^N g(f(X_i), Y_i)(z) + \frac{2}{N} \mathrm{E}_{\sigma_i}\left[ \sup_{f \in \mathcal{F}} \sum_{i=1}^N g(f(X_i)(z), Y_i(z))\sigma_i \right] + 3\sqrt{\frac{\log 1/\delta}{N}}.$$

*Here, $\mathrm{E}_{\sigma_i}[\sup_{f \in \mathcal{F}} \sum_{i=1}^N g(f(X_i)(z), Y_i(z))\sigma_i] = \int_\Omega \sup_{f \in \mathcal{F}} \sum_{i=1}^N g(f(X_i)(z), Y_i(z))\sigma_i(\omega)\mathrm{d}\mu(\omega)$.*

**Proof** With probability at least $1 - \delta$, we have

$$\mathrm{E}[g(f(X), Y)](z) = \mathrm{E}[g(f(X)(z), Y(z))]$$

$$\le \frac{1}{N} \sum_{i=1}^N g(f(X_i)(z), Y_i(z)) + \frac{2}{N} \mathrm{E}_{\sigma_i}\left[ \sup_{f \in \mathcal{F}} \sum_{i=1}^N g(f(X_i)(z), Y_i(z))\sigma_i \right] + 3\sqrt{\frac{\log 1/\delta}{N}}$$

$$= \frac{1}{N}\sum_{i=1}^{N} g(f(X_i), Y_i)(z) + \frac{2}{N}\mathrm{E}_{\sigma_i}\left[\sup_{f \in \mathcal{F}} \sum_{i=1}^{N} g(f(X_i)(z), Y_i(z))\sigma_i\right] + 3\sqrt{\frac{\log 1/\delta}{N}}.$$

The inequality follows by Theorem 3.1 in Mohri et al. (2012). $\qquad\square$

According to Hashimoto et al. (2023a), the $\mathcal{A}$-valued Rademacher complexity for the RKHM associated with an $\mathcal{A}$-valued kernel $k$ is upperbounded by $\|k(x_i, x_i)\|_{\mathcal{A}}$ as follows.

**Lemma E.4** *Let $B > 0$ and $\mathcal{F} = \{f \in \mathcal{M}_k \mid \|f\|_k \leq B\}$. Then, we have*

$$\hat{R}_{\mathcal{A},N}(\mathbf{x}, \mathcal{F}) \leq_{\mathcal{A}} \frac{B}{N}\left(\sum_{i=1}^{N} \|k(x_i, x_i)\|_{\mathcal{A}}\right)^{1/2} 1_{\mathcal{A}}.$$

Combining these results with the following lemma completes the proof of Theorem 4.1.

**Lemma E.5** *Let $k_n$ be $k_n^{\mathrm{poly},q}$, $\hat{k}_n^{\mathrm{prod},q}$, or $k_n^{\mathrm{sep},q}$ defined in Section 3. Then, for $x \in \mathcal{A}_0^d$, $D(k_n, x) \leq D(k_{n+1}, x)$.*

**Proof** We evaluate $\|k_n(x, x)\|_{\mathcal{A}}$. For $k_n^{\mathrm{poly},q}$, we have

$$\|k_n^{\mathrm{poly},q}(x, x)\| = \left\|S_n\left(\sum_{i=1}^{d}\alpha_i(R_n(x_i)^*)^q R_n(x_i)^q\right)\right\|_{\mathcal{A}} \leq \left\|\sum_{i=1}^{d}\alpha_i(R_n(x_i)^*)^q R_n(x_i)^q\right\|_{\mathrm{op}}$$

$$\leq \sum_{i=1}^{d}\alpha_i\|R_n(x_i)\|_{\mathrm{op}}^{2q}.$$

The first inequality is derived by the following inequalities for $A \in \mathbb{C}^{n \times n}$:

$$|S_n(A)(z)| = \left|\frac{1}{n}\sum_{j,l=0}^{n-1} A_{j,l}\mathrm{e}^{\mathrm{i}(j-l)z}\right|$$

$$= \left|\mathrm{tr}\left(\frac{1}{n}\mathbf{1}\,\mathrm{diag}(\mathrm{e}^0, \dots, \mathrm{e}^{\mathrm{i}(n-1)})A\,\mathrm{diag}(\mathrm{e}^{0z}, \dots, \mathrm{e}^{\mathrm{i}(n-1)z})\right)\right|$$

$$\leq \frac{1}{n}\|\mathbf{1}\|_1\|\,\mathrm{diag}(\mathrm{e}^0, \dots, \mathrm{e}^{\mathrm{i}(n-1)})A\,\mathrm{diag}(\mathrm{e}^{0z}, \dots, \mathrm{e}^{\mathrm{i}(n-1)z})\|_{\mathrm{op}} \leq \|A\|_{\mathrm{op}}$$

for any $z \in \mathbb{T}$. Here, $\mathrm{tr}(A)$ is the trace of a matrix $A$, $\mathrm{diag}(a_1, \dots, a_n)$ for $a_1, \dots, a_n \in \mathbb{C}$ is the diagonal matrix whose diagonal elements are $a_1, \dots, a_n$, $\mathbf{1} \in \mathbb{C}^{n \times n}$ is the matrix whose elements are all 1, and $\|A\|_1$ for $A \in \mathbb{C}^{n \times n}$ is the trace norm defined as $\|A\|_1 = \mathrm{tr}((A^*A)^{1/2})$. We used the inequality $|\mathrm{tr}(AB)| \leq \|A\|_1\|B\|_{\mathrm{op}}$ for $A, B \in \mathbb{C}^{n \times n}$ (Conway, 2007, Chapter IX, Section 2).

In addition, there exists $v \in \mathbb{C}^{n \times n}$ such that $\|R_n(x)\|_{\mathrm{op}} = \|R_n(x)v\|$ and $\|v\| = 1$. Thus, we have

$$\|R_n(x)\|_{\mathrm{op}} = \|R_n(x)v\| = \|Q_n^* M_x Q_n v\| = \left\|Q_n^* M_x Q_{n+1}\begin{bmatrix}v\\0\end{bmatrix}\right\| \leq \left\|Q_{n+1}^* M_x Q_{n+1}\begin{bmatrix}v\\0\end{bmatrix}\right\|$$

$$\leq \|Q_{n+1}^* M_x Q_{n+1}\|_{\mathrm{op}} = \|R_{n+1}(x)\|_{\mathrm{op}},$$

where $Q_n = [e_1, \dots, e_n]$ and $e_j(z) = \mathrm{e}^{\mathrm{i}jz}$ is the Fourier function. $\qquad\square$

## F  RELATIONSHIP BETWEEN $q$ AND THE INTERACTIONS

Theorem 3.4 implies that the interactions along $z \in \mathbb{T}$ in the kernels become small as $n$ grows. On the other hand, we also have the parameter $q$ that describes how many Toeplitz matrices created by $R_n$ are involved in the kernel. Regarding the relationship between $q$ and the interactions, we have the following remark.

**Remark F.1** *We expect that the convergence in Theorem 3.4 becomes slower as $q$ becomes larger, and the interaction along $z \in \mathbb{T}$ becomes larger. This is based on the observation that the value related to the Fejér kernel on $\mathbb{T}^q$ is described by the corresponding value related to the Fejér kernel on $\mathbb{T}^{q-1}$, especially in the case of $q = 2$.*

To understand the relationship between $q$ and the interactions, we observe the convergence shown in Lemma 3.6 from the perspective of $q$. The following lemma is by Brandlini and Travaglini Brandolini & Travaglini (1997).

**Lemma F.2** *Let $H_n^{q,P}(t) = n^q \int_0^1 \int_{\mathbb{R}^q} \chi_{jP}(r) \mathrm{e}^{\mathrm{i} n r \cdot t} \mathrm{d}r \mathrm{d}j$ for $t \in \mathbb{T}^q$, where $\chi_S$ is the characteristic function with respect to a set $S$. Then, we have $F_n^{q,P}(t) = \sum_{m \in \mathbb{Z}^q} H_n^{q,P}(t + 2\pi m)$.*

**Proof** Let $u_{j,t}(r) = \chi_{jP}(r) \mathrm{e}^{\mathrm{i} r \cdot t}$. Then, we have

$$F_n^{q,P}(t) = \frac{1}{n} \sum_{j=1}^{n-1} \sum_{r \in jP \cap \mathbb{Z}^q} \mathrm{e}^{\mathrm{i} r \cdot t} = \frac{1}{n} \int_0^n \sum_{r \in \mathbb{Z}^q} \chi_{jP}(r) \mathrm{e}^{\mathrm{i} r \cdot t} \mathrm{d}j = \frac{1}{n} \int_0^n \sum_{r \in \mathbb{Z}^q} u_{j,t}(r) \mathrm{d}j$$

$$= \frac{1}{n} \int_0^n \sum_{m \in \mathbb{Z}^q} \widehat{u_{j,t}}(m) \mathrm{d}j = \sum_{m \in \mathbb{Z}^q} \frac{1}{n} \int_0^n \int_{\mathbb{R}^q} \chi_{jP}(r) \mathrm{e}^{\mathrm{i} r \cdot t} \mathrm{e}^{2\pi \mathrm{i} r \cdot m} \mathrm{d}r \mathrm{d}j$$

$$= \sum_{m \in \mathbb{Z}^q} \int_0^1 \int_{\mathbb{R}^q} \chi_{jnP}(r) \mathrm{e}^{\mathrm{i} r \cdot t} \mathrm{e}^{2\pi \mathrm{i} r \cdot m} \mathrm{d}r \mathrm{d}j = \sum_{m \in \mathbb{Z}^q} \int_0^1 \int_{\mathbb{R}^q} \chi_{jP}\left(\frac{r}{n}\right) \mathrm{e}^{\mathrm{i} r \cdot t} \mathrm{e}^{2\pi \mathrm{i} r \cdot m} \mathrm{d}r \mathrm{d}j$$

$$= n^q \sum_{m \in \mathbb{Z}^q} \int_0^1 \int_{\mathbb{R}^q} \chi_{jP}(r) \mathrm{e}^{\mathrm{i} n r \cdot t} \mathrm{e}^{2\pi \mathrm{i} n r \cdot m} \mathrm{d}r \mathrm{d}j,$$

where $\hat{\cdot}$ is the Fourier transform, and the forth equality is by the Poisson summation formula. □

To understand the convergence, we split $F_n^{q,P}(t) = \sum_{m \in \mathbb{Z}^q} H_n^{q,P}(t + 2\pi m)$ into two parts: $H_n^{q,P}(t)$ and $\tilde{H}_n^{q,P}(t) := \sum_{m \neq 0} H_n^{q,P}(t + 2\pi m)$. In the convergence discussed in Lemma 3.6, the convolution $g * H_n^{q,P}(z)$ goes to $g(z)$. On the other hand, $g * \tilde{H}_n^{q,P}(z)$ goes to 0 (Brandolini & Travaglini, 1997, Theorem 1). Regarding the convergence of $g * H_n^{q,P}(z)$ we can evaluate it as follows, especially for $q = 2$.

**Lemma F.3** *For $z \in \mathbb{T}$, we have*

$$\lim_{n \to \infty} g * H_n^{q,P}(z) - g(z) = \int_0^1 \lim_{n \to \infty} \left( \int_{t \in \mathbb{T}^q} \int_{jnP} \mathrm{e}^{\mathrm{i} r \cdot (z-t)} g(t) \mathrm{d}r \mathrm{d}t - g(z) \right) \mathrm{d}j.$$

*Moreover, let $h_s^P(z) = \int_{t \in \mathbb{T}^q} \int_{sP} \mathrm{e}^{\mathrm{i} r \cdot (z-t)} g(t) \mathrm{d}r \mathrm{d}t - g(z)$. Let $q = 2$, $Q = \{(r_1, -r_2) \mid (r_1, r_2) \in P\}$, $P_1' = [-1, 1]$, and $P_2 = \{(r_1, r_2) \mid |r_1| \leq 1, |r_2| \leq 1, |r_1 + r_2| \leq 1, |r_1 - r_2| \leq 1\}$. Let $g(z) = \overline{x_1(z_1) x_2(z_2)} y_1(z_3) y_4(z_4)$ for $z = [z_1, z_2, z_3, z_4]$. Then, for $z \in \mathbb{T}^2$, there exists $C(z) \geq 0$ such that we have*

$$\frac{1}{2}|h_{jn}^P(z) + h_{jn}^Q(z)| \leq \frac{1}{2}C(z) \sum_{j=1}^2 \left| \int_{t \in \mathbb{T}} \int_{jnP_1'} \mathrm{e}^{\mathrm{i} r(z_1 - t)} x_j(t_j) \mathrm{d}r \mathrm{d}t - x_j(z_j) \right|$$

$$+ \frac{1}{2} \left| \int_{\mathbb{T}^2} \int_{jnP_2} \mathrm{e}^{\mathrm{i} r \cdot (z-t)} g(t) \mathrm{d}r \mathrm{d}t_1 - g(z) \right|. \tag{8}$$

**Proof** We have

$$n^q \int_{t \in \mathbb{T}^q} \int_0^1 \int_{jP} \mathrm{e}^{\mathrm{i} n r \cdot (z-t)} g(t) \mathrm{d}r \mathrm{d}j \mathrm{d}t - g(z) = \int_0^1 \int_{t \in \mathbb{T}^q} \int_{jnP} \mathrm{e}^{\mathrm{i} r \cdot (z-t)} g(t) \mathrm{d}r \mathrm{d}t - g(z) \mathrm{d}j.$$

Let $h_s^P(z) = \int_{t \in \mathbb{T}} \int_{sP} \mathrm{e}^{\mathrm{i} r \cdot (z-t)} g(t) \mathrm{d}r \mathrm{d}t - g(z)$. Then, $h_0^P(z) = 0$, and since $\lim_{s \to \infty} h_s^P(z) = 0$, there exists $D(z) > 0$ such that $|h_s^P(z)| \leq D(z)$ for any $s > 0$. Thus, $|h_{jn}^P(z)| \leq D(z)$ for any

$n \in \mathbb{N}$ and $0 < j \le 1$. Thus, by the bounded convergence theorem, we have

$$\lim_{n \to \infty} \int_0^1 \int_{t \in \mathbb{T}^q} \int_{jnP} \mathrm{e}^{\mathrm{i}r \cdot (z-t)} g(t) \mathrm{d}r \mathrm{d}t - g(z) \mathrm{d}j = \int_0^1 \lim_{n \to \infty} \int_{t \in \mathbb{T}^q} \int_{jnP} \mathrm{e}^{\mathrm{i}r \cdot (z-t)} g(t) \mathrm{d}r \mathrm{d}t - g(z) \mathrm{d}j.$$

In addition, let $P_1 = [-1, 1]^2$ and $P_2 = \{(r_1, r_2) \mid |r_1| \le 1, |r_2| \le 1, |r_1 + r_2| \le 1, |r_1 - r_2| \le 1\}$. Then, we have

$$h_{jn}^P(z) + h_{jn}^Q(z) = h_{jn}^{P_1}(z) + h_{jn}^{P_2}(z).$$

For $h_{jn}^{P_1}(z)$, we have

$$|h_{jn}^{P_1}(z)| = \left| \int_{\mathbb{T}} \int_{-jn}^{jn} \mathrm{e}^{\mathrm{i}r(z_1-t)} x_1(t) \mathrm{d}r \mathrm{d}t \int_{\mathbb{T}} \int_{-jn}^{jn} \mathrm{e}^{\mathrm{i}r(z_2-t)} x_2(t) \mathrm{d}r \mathrm{d}t - g(z) \right|$$

$$\le \left| \int_{\mathbb{T}} \int_{-jn}^{jn} \mathrm{e}^{\mathrm{i}r(z_1-t)} x_1(t) \mathrm{d}r \mathrm{d}t \left( \int_{\mathbb{T}} \int_{-jn}^{jn} \mathrm{e}^{\mathrm{i}r(z_2-t)} x_2(t) \mathrm{d}r \mathrm{d}t - x_2(z_2) \right) \right|$$

$$+ \left| x_2(z_2) \left( \int_{\mathbb{T}} \int_{-jn}^{jn} \mathrm{e}^{\mathrm{i}r(z_1-t)} x_1(t) \mathrm{d}r \mathrm{d}t - x_1(z_1) \right) \right|.$$

In the same manner as $h_s^P(z)$, there exists $\tilde{C}(z) > 0$ such that $|\int_{\mathbb{T}} \int_{-jn}^{jn} \mathrm{e}^{\mathrm{i}r(z_1-t)} x_1(t) \mathrm{d}r \mathrm{d}t| \le \tilde{C}(z)$. By setting $C(z) = \max\{\tilde{C}(z), |x_2(z_2)|\}$, we obtain the result. $\qquad\square$

The left hand side is the average of $h_{jn}^P(z)$ and $h_{jn}^Q(z)$, where $P$ and $Q$ are symmetric with respect to the second coordinate $r_2$. The first term in the right hand side of Eq. (8) is described by the convergence of $h_{jn}^{[-1,1]}(z)$, which corresponds to the one-dimensional $((q-1)$-dimensional) Fejér kernel. Since we have an additional term $1/2| \int_{\mathbb{T}^2} \int_{jnP_2} \mathrm{e}^{\mathrm{i}r \cdot (z-t)} g(t) \mathrm{d}r \mathrm{d}t - g(z)|$, if $C(z) > 1$, the convergence of $h_{jn}^P(z)$ is expected to be slower than its one-dimensional counterpart $h_{jn}^{[-1,1]}(z)$. Based on this observation, we expect that the convergence in Theorem 3.4 becomes slower as $q$ becomes larger, and the interaction along $z \in \mathbb{T}$ becomes larger.

## G  REPRESENTER THEOREM FOR RKHMs OVER GENERAL $C^*$-ALGEBRAS

Hashimoto et al. (Hashimoto et al., 2023b, Proposition 4.5) showed an approximate representer theorem for RKHMs over von Neumann-algebras. Since $C(\mathbb{T})$ is not a von Neumann-algebra, we generalize the theorem to that for general $C^*$-algebras as follows.

**Proposition G.1** *Let $x_1, \ldots, x_N \in \mathcal{A}^d$. Let $g : \mathcal{A}^N \to \mathcal{A}_+$ be a continuous map, let $h : \mathcal{A}_+ \to \mathcal{A}_+$ satisfy $h(c) \le_{\mathcal{A}} h(d)$ for $c, d \in \mathcal{A}_+$ with $c \le_{\mathcal{A}} d$. Let $L(f) = g(f(x_1), \ldots, f(x_N)) + h(|f|_k)$ for $f \in \mathcal{M}_k$. If there exists a solution $f_0$ of the minimization problem $\min_{f \in \mathcal{M}_k} L(f)$, then for any $\epsilon > 0$, there exists $\tilde{f}$ that admits the representation of $\tilde{f} = \sum_{i=1}^N \phi(x_i) c_i$ and $\|L(\tilde{f}) - L(f_0)\| \le \epsilon$.*

**Proof** For $f_1, f_2 \in \mathcal{M}_k$, let $\theta_{f_1, f_2} : \mathcal{M}_k \to \mathcal{M}_k$ defined as $\theta_{f_1, f_2}(v) = f_1 \langle f_2, v \rangle_k$ for $v \in \mathcal{M}_k$. Let $\mathcal{N}_k$ be the submodule generated algebraically by $\{\phi(x_1), \ldots, \phi(x_N)\}$, let $\mathcal{B}_0 = \mathrm{Span}\{\theta_{f_1, f_2} \mid f_1, f_2 \in \mathcal{N}_k\}$, and let $\mathcal{B} = \overline{\mathcal{B}_0}$. In addition, let $\mathcal{L}(\mathcal{M}_k)$ be the (unital) $C^*$-algebra of adjointable $\mathcal{A}$-linear operators on $\mathcal{M}_k$ (see (Lance, 1995, p8)). Since $\mathcal{B}$ is a $C^*$-subalgebra of $\mathcal{L}(\mathcal{M}_k)$ (Note $\theta_{f_1, f_2} \theta_{f_3, f_4} = \theta_{f_1, f_4 \langle f_3, f_2 \rangle}$ and $\theta_{f_1, f_2}^* = \theta_{f_2, f_1}$), $\mathcal{B}_0$ has a net $\{b_i\}_i$ such that $0 \le_{\mathcal{L}(\mathcal{M}_k)} b_i \le_{\mathcal{L}(\mathcal{M}_k)} 1$ and $\lim_i \theta_{f_1, f_2} b_i = \lim_i b_i \theta_{f_1, f_2} = \theta_{f_1, f_2}$ for any $f_1, f_2 \in \mathcal{N}_k$ (Davidson, 1996, Theorem I.4.8). For each $v \in \mathcal{N}_k$, we have

$$\|v - b_i v\|_k^2 = \| \langle (1 - b_i)v, (1 - b_i)v \rangle_k \|_{\mathcal{A}} = \|\theta_{(1-b_i)v, (1-b_i)v}\|_{\mathcal{L}(\mathcal{M}_k)}$$

$$= \|(1 - b_i) \theta_{v,v} (1 - b_i)^*\|_{\mathcal{L}(\mathcal{M}_k)} = \|(1 - b_i) \theta_{v,v} (1 - b_i)\|_{\mathcal{L}(\mathcal{M}_k)}.$$

Thus, we have $\lim_i b_i v = v$. Since $g$ is continuous, we have

$$\lim_i g(\langle \phi_k(x_1), b_i f_0 \rangle_k, \ldots, \langle \phi_k(x_N), b_i f_0 \rangle_k) = \lim_i g(\langle b_i^* \phi_k(x_1), f_0 \rangle_k, \ldots, \langle b_i^* \phi_k(x_N), f_0 \rangle_k)$$

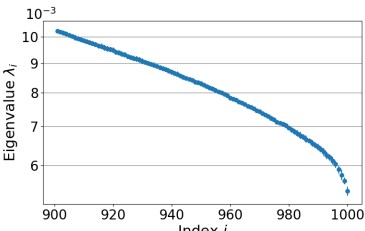

Figure 6: Eigenvalues of the Gram matrix $\mathbf{G}(0)$ indexed as the descending order for the regression task. (Average value of results of five different runs. The error bar represents the standard deviation.)

$$= \lim_i g(\langle b_i \phi_k(x_1), f_0 \rangle_k, \ldots, \langle b_i \phi_k(x_N), f_0 \rangle_k) = g(\langle \phi_k(x_1), f_0 \rangle_k, \ldots, \langle \phi_k(x_N), f_0 \rangle_k).$$

Therefore, for $\epsilon > 0$, there exists $i$ such that

$$\|g((b_i f_0)(x_1), \ldots, (b_i f_0)(x_N)) - g(f_0(x_1), \ldots, f_0(x_N))\|_{\mathcal{A}} \leq \epsilon.$$

Since $b_i \geq_{\mathcal{L}(\mathcal{M}_k)} 0$, there exists $c \geq_{\mathcal{L}(\mathcal{M}_k)} 0$ such that $b_i = c^2$. Thus, $b_i - b_i^2 = c^2 - c^4 = c(1 - c^2)c = c^*(1 - b_i)c \geq_{\mathcal{L}(\mathcal{M}_k)} 0$. Thus, $0 \leq_{\mathcal{L}(\mathcal{M}_k)} b_i^2 \leq_{\mathcal{L}(\mathcal{M}_k)} b_i \leq_{\mathcal{L}(\mathcal{M}_k)} 1$, and we have $|b_i f_0|_k^2 = \langle f_0, b_i^2 f_0 \rangle_k \leq_{\mathcal{A}} \langle f_0, f_0 \rangle_k = |f_0|_k$. As a result, we obtain

$$\begin{aligned} 0 &\leq_{\mathcal{A}} L(b_i f_0) - L(f_0) \\ &= g((b_i f_0)(x_1), \ldots, (b_i f_0)(x_N)) + h(|b_i f_0|_k) - g(f_0(x_1), \ldots, f_0(x_N)) - h(|f_0|_k) \\ &\leq_{\mathcal{A}} g((b_i f_0)(x_1), \ldots, (b_i f_0)(x_N)) - g(f_0(x_1), \ldots, f_0(x_N)). \end{aligned}$$

Since $b_i f_0 \in \mathcal{N}_k$, setting $\tilde{f} = b_i f_0$ completes the proof of the proposition. $\square$

## H  GENERALIZATION TO OTHER $C^*$-ALGEBRAS

Setting $R_n$ and $S_n$ for more general $C^*$-algebras has been investigated. Using these results and replacing $R_n$ and $S_n$ in Definition 3.2, we can define positive definite kernels for the $\mathcal{A}^d$-valued inputs and $\mathcal{A}$-valued output for a more general $C^*$-algebra $\mathcal{A}$.

**Continuous functions on high-dimensional torus (Leimbach & van Suijlekom, 2024)**  For $\mathcal{A} = C(\mathbb{T}^m)$, let $e_j(z) = e^{ij \cdot z}$ for $j \in \mathbb{Z}^m$ and $B_n = \{j \in \mathbb{Z}^m \mid \|j\| \leq n\}$, and consider the space $\mathrm{Span}\{e_j \mid j \in B_n\}$. Here, $\|\cdot\|$ is the Euclidean norm. We consider generalized matrices whose elements are indexed by $\mathbb{Z}^m$, and set $R_n(x) = (\int_{\mathbb{T}^m} x(t) e^{-i(j-l) \cdot t} dt)_{j,l \in B_n}$ and $S_n(A)(z) = (1/|B_n|) \sum_{j,l \in B_n} A_{j,l} e^{i(j-l)z}$.

**Continuous functions on the sphere (Rieffel, 2004)**  For $\mathcal{A} = C(S^2)$, let $\rho_n$ be the $n$-dimensional irreducible group representation of $SU(2)$ (special unitary group of degree 2) and let $V_n$ be the representation space. Let $P \in \mathcal{B}(V_n)$ be a projection, where $\mathcal{B}(V_n)$ is the space of bounded linear operators on $V_n$. We set $R_n(x) = n \int_{SU(2)} x(g) \rho_n(g)^* P \rho_n(g) dg$ and $S_n(A)(g) = \mathrm{tr}(A \rho_n(g)^* P \rho_n(g))$.

## I  EXPERIMENTAL DETAILS AND ADDITIONAL RESULTS

We provide experimental details below. All the experiments in this paper were executed with Python 3.9 on an Intel(R) Core(TM) i9-10885H 2.4GHz processor with the Windows 10 operating system.

### I.1  EXPERIMENT WITH SYNTHETIC DATA

We estimated $f$ using kernel ridge regression with the regularization parameter $\lambda = 0.01$. We generated 1000 input test samples in the same manner as the input training samples, and evaluated

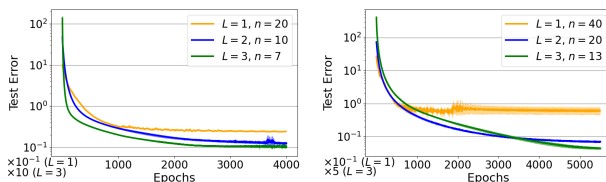

Figure 7: Test error of the regression task with deep approach with different $n$ and $L$.

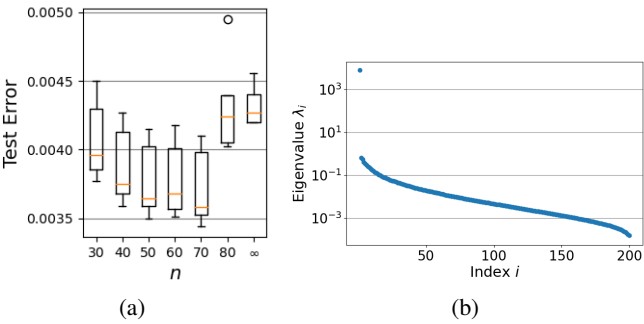

(a)             (b)

Figure 8: (a) Test error of the image recovering task with different values of $n$ and the kernel $k = \hat{k}_n^{\mathrm{prod},q}$. (Box plot of results of five independent runs with different training and test data.) (b) Eigenvalues of the Gram matrix $\mathbf{G}(0)$ indexed as the descending order for the image recovering task. (Average value of results of five different runs. The error bar represents the standard deviation.)

the test error $1/N \sum_{i=1}^{N} \|\hat{f}(\tilde{x}^i) - \tilde{y}^i\|_{L^2(\mathbb{T})}$, where $\hat{f}$ is the estimation of $f$ and $\tilde{x}^i$ and $\tilde{y}^i$ are input and output test samples, respectively. For $x^i$, $y^i$, $\tilde{x}^i$, and $a$, we discretized each function with 30 equally spaced points on $\mathbb{T}$. In addition, to investigate the positive definiteness of $\hat{k}_n^{\mathrm{prod},q}$ discussed in Remark 3.10, we computed the eigenvalues of the Gram matrix $\mathbf{G}(0)$ at 0. Figure 6 shows the result. The index $i$ is determined in the descendent order with respect to the value of $\lambda_i$. Thus, we can see that although we set $\beta_n$ as a small value, all the eigenvalues of $\mathbf{G}(0)$ are positive in this case.

We also show additional results regarding the deep approach in Figure 7. The result is the average $\pm$ the standard deviation of three independent runs. We set different values of $n$ with the same deep setting considered in Subsection 7.1.

## I.2   Experiment with MNIST

We generated 200 input test samples in the same manner as the input training samples, and evaluated the test error $1/N \sum_{i=1}^{N} \|\hat{f}(\tilde{x}^i) - \tilde{y}^i\|_{L^2(\mathbb{T})}$, where $\hat{f}$ is the estimated map that maps the image with the missing part to its original image, and $\tilde{x}^i$ and $\tilde{y}^i$ are input and output test samples, respectively. Figure 8 (a) shows the test error. We can see that the test error becomes the smallest when $n = 70$, but it becomes large when $n$ is smaller or larger than 70. In addition, to investigate the positive definiteness of $\hat{k}_n^{\mathrm{prod},q}$, we computed the eigenvalues of the Gram matrix $\mathbf{G}(0)$ at 0. Figure 8 (b) shows the result. The index $i$ is determined in the descendent order with respect to the value of $\lambda_i$. Thus, we can see that although we set $\beta_n$ as a small value, all the eigenvalues of $\mathbf{G}(0)$ are positive in this case.

