# OpenReview forum: "Spectral Truncation Kernels: Noncommutativity in $C^*$-algebraic Kernel Machines"
_ICLR.cc/2025/Conference — Submitted to ICLR 2025_

### Official Review · Reviewer_jZ5N · 2024-11-01

**Soundness:** 3
**Presentation:** 3
**Contribution:** 3
**Rating:** 6
**Confidence:** 2

**Summary:**

In this paper, authors propose a set of positive definite spectral truncation kernels, which is a class of the $C^*$-Algebra-valued kernel. The definitions of the proposed kernels involve several concepts, including $C^*$-Algebra, function-valued kernel, spectral truncation, and the torus. The authors provide a theoretical analysis of the convergence and generalization.  In addition, the authors introduce a noncommutativity and further illustrate its effectiveness with numerical results.

**Strengths:**

1. This paper is well written and organized.
2. Theoretical analysis is detailed, enabling the proposed method to have solid theoretical support.
3. The authors consider the deep spectral truncation kernel, sharing the advantage of both deep model and spectral truncation kernel, to improve the representation power.

**Weaknesses:**

Please see the **Questions** section.

**Questions:**

1. In line 156, $z \in \mathbb{T}$ be the Fourier function. Why $z$ is a function? Based on the **Example 2.2**, $\mathbb{T}$ is a set, and the elements of $\mathbb{T}$ are real numbers. This puzzled me.
2. One of the main contributions of this paper is that the authors generalize typical kernels by introducing the noncommutativity of the products appearing in the kernels and showing their advantages. This is because $\mathcal{R}_n (x)$ and $\mathcal{R}_n (y)$, based on the spectral truncation, is noncommutative. However, the benefits of introducing noncommutativity in terms of convergence and generalization were not found. The effect of the noncommutativity is just illustrated in the experiment part. Or am I missing some details?
3. **Theory 3.4** gives the theoretical result of the convergence of proposed kernels. Does this mean that the proposed $k_n^{poly,q}(x, y)(z)$, $k_n^{prod,q}(x, y)(z)$, and $k_n^{seq,q}(x, y)(z)$  can approximate $k^{poly,q}(x, y)(z)$, $k^{prod,q}(x, y)(z)$, and $k^{seq,q}(x, y)(z)$, respectively? So is there any theoretical guidance for the selection of $n$, that is, how much $n$ can be well approximated?
4. From **Theory 4.1**, we can observe that the generalization bound is related to the trace of the kernel. How is this different from the previous theoretical results?
5. $n$ is the number of orthogonal bases. Therefore, the complexity of the model is larger if $n$ is larger, and the representation power of the model is better. This phenomenon also occurs in the general learning process or kernel function approach strategies. How is this different from them?
6. To obtain $c(z)$, computing $(G(z)+\lambda I)^{-1}y(z)$. Poor scalability.
7. Some definitions are in the complex number domain, while some are in the real number domain. It is confusing for me. When to do it in the complex number domain and when to do it in the real number domain. It can include a pseudo-code to show the details.

---

> ### Author Response · Authors · 2024-11-20
>
> Thank you very much for your constructive comments. We addressed you comments and questions and revised the paper. We summarize the answers below.
>
> **[Q1] Presentation regarding the Fourier function**
> We are sorry for the confusion.
> We replaced the sentence with "Let $e_j$ be the Fourier function defined as $e_j(z)=\mathrm{e}^{\mathrm{i}jz}$ for $j\in\mathbb{Z}$".
>
> **[Q2] Benefits of introducing noncommutativity**
> Although benefits in the terms of convergence and generalization are also important, but the main scope of this paper is to propose kernels **going beyond commutative and separable kernels with lower computational cost** than existing operator-valued kernels with the framework of vvRKHSs.
> Separable and commutative kernels are two extreme cases regarding the dependencies between input and output variables.
> For separable kernels, the output is determined only by the global information of the input.
> On the other hand, commutative kernels only identify the pointwise (completely local) dependencies.
> **The proposed kernel fill a gap between separable and commutative kernels** with lower computational cost than existing operator-valued kernels.
> Indeed, the proposed kernel is indexed by the truncation parameter $n$, and we showed that the proposed kernels converge to the commutative kernels as $n$ goes to infinity.
> On the other hand, if $n=1$, then the proposed kernels are separable kernel.
> **If $n$ is small, the proposed kernel focuses more on global information of input functions.
> If $n$ is large, the proposed kernel focuses more on local information of input functions.**
> This feature is achieved by introducing the noncommutativity to typical commutative kernels by applying the theory of spectral truncation, and is described by the property of the Fejer kernel.
> We revised Sections 1 and 3 and added Appendices A and B to clarify the benefit of introducing the noncommutativity.
>
> **[Q3] Theoretical guidance for the selection of $n$, how much $n$ is suitable for well approximated**
> Although $k_n^{\operatorname{poly},q}$, $k_n^{\operatorname{prod},q}$, and $k_n^{\operatorname{sep},q}$ can approximate $k^{\operatorname{poly},q}$, $k^{\operatorname{prod},q}$, and $k^{\operatorname{sep},q}$, our goal is not approximating the commutative kernels $k^{\operatorname{poly},q}$, $k^{\operatorname{prod},q}$, and $k^{\operatorname{sep},q}$.
> As we explained above, by setting $n$ as a finite value, we can go beyond separable and commutative kernels.
> Indeed, the parameter $n$ controls the input and output dependencies.
> If $n$ is small, the proposed kernel focuses more on global information of input functions.
> If $n$ is large, the proposed kernel focuses more on local information of input functions.
> This feature is described by the property of the Fejer kernel.
> We can determine optimal $n$ in the sense of the dependencies by observing the Fejer function $F_n^{2q,P}$.
> Since the proposed kernel is defined by the convolution of the input function with the Fejer kernel, the volume of the region where the value of the Fejer kernel is sufficiently large corresponds to the range of local dependencies.
> Thus, if we have a information of the local dependencies, then we can choose $n$ based on the values of $F_n^{2q,P}$.
>
> In Appendix B, we added Remark B.1 about determining $n$ discussed above. We also added Figure 2 for helping understand the property of the Fejer kernel.
>
> **[Q4,Q5] Defference from previous results of generalization bounds**
> The argument regarding the representation power and model complexity is common.
> However, since we proposed a new kernel, it is important to check this common property is also valid to the proposed kernel.
> This paper is the first step of proposing kernels that beyond separable and commutative kernels based on the noncommutativity and speatral truncation.
> More detailed analysis of what benefit is obtaind in the sense of generalization bound compared to other machine learning methods is future work.

---

> > ### Author Response · Authors · 2024-11-20
> >
> > **[Q6] Scarability of computing $\mathbf{c}(z)$**
> > The computation of $(\mathbf{G}(z)+\lambda I)^{-1}\mathbf{y}(z)$ is for the most standard case for obtaining $\mathbf{c}(z)$ without any approximation technique.
> > We emphasize that for the case without approximations, the computational cost is lower than vvRKHS methods with existing nonseparable operator-valued kernels such as transformable kernels.
> > As we stated in the last paragraph of Section 5, for the proposed kernel with the framework of RKHMs, the computational cost for obtaining $\mathbf{c}(z_1),\ldots,\mathbf{c}(z_m)$ for different points $z_1,\ldots,z_m$ is $O((q+m)n^2N^2+mN^3)$.
> > On the other hand, that for existing nonseparable operator-valued kernels with the framework of vvRKHSs is $O(m^3N^3)$.
> > Thus, if $(q+m)n^2<m^3N$, then the proposed kernels are more computationally efficient than the existing kernels, such as transformable kernels.
> > In addition, we can apply low-rank approximation such as the Nystrom method to each $\mathbf{G}(z_i)\in\mathbb{C}^{N\times N}$.
> > In this way, we can reduce the computational cost with respect to $N$ for the proposed kernel.
> > We revised the last paragraph of Section 5 to clarify the above point.
> >
> > **[Q7] Complex-valued and real-valued notions**
> > We are sorry for the confusion.
> > In practical applications, we can use complex-valued kernels.
> > However, only when we obtain the generalization bound, we need to restrict the kernels to be real-valued.
> > This is because the theory of generalization bound is for real-valued functions in general.
> > We added the explanation that we consider real-valued kernels for the generalization bound analysis before Theorem 4.1.

---

> > > ### Comment · Reviewer_jZ5N · 2024-11-21
> > > **Comment**
> > >
> > > Thank you for your response. It has deepened my understanding of this work. I am considering improving my score. But, I still have a problem. As maintained above, the proposed method indeed reduces computational cost. However, it still needs $\mathcal{O}(N^3)$, which is not acceptable for large-scale data tasks. DNN can also capture the global and local features and has a better representation capability. What are the benefits of the proposed method? Or, what is the necessity of studying this method?

---

> > > > ### Author Response · Authors · 2024-11-21
> > > >
> > > > Thank you very much for your additional questions, and thank you for considering improving the score.
> > > >
> > > > We have the following three points regarding the advantages of applying the proposed kernels over applying DNNs.
> > > > - In many cases, kernel methods performs better than DNN when the number of samples is limited [1]. For large-scale data tasks, as we discussed in the answer of [Q6], we can use approximation methods such as the Nystrom method to reduce the computational cost with respect to $N$.
> > > > - One potential application of the proposed kernel is the application to operator learning (Please see Appendix D for more details). In [2], the authors show the kernel method for operator learning either matches or beats the performance of NN-based methods on a majority of benchmarks. Thus, if we apply the proposed kernel to operator learning, its performance can be better than NN-based methods.
> > > > - An advantage of kernel methods over DNNs is its theoretical solidness. As the authors of [2] also insist in their paper, with kernel methods, we have convergence guarantees, a priori error estimates, Bayesian uncertainty quantification, and so on.
> > > >
> > > > [1] Tingting Wang, Huayou Su, and Junbao Li, "DWS-MKL: Depth-width-scaling multiple kernel learning for data classification", Neurocomputing, 411: 455-467, 2020.
> > > > [2] Pau Batlle, Matthieu Darcy, Bamdad Hosseini, and Houman Owhadi, "Kernel methods are competitive for operator learning", Journal of Computational Physics, 496(1):112549, 2024.

---

> > > > > ### Comment · Reviewer_jZ5N · 2024-11-22
> > > > > **Comment**
> > > > >
> > > > > Thanks for your response. We acknowledge that kernel methods are better suited for situations with a small sample size compared to deep networks. The reference [1] is the deep kernel machine, which combines the deep network with the kernel method. So, in a sense, does it borrow the hierarchical structure of deep networks? In that case, would it tend to be more suitable for large-scale data? And, in the proposed method, the authors also extend the proposed method to the deep case. If yes, what are the benefits of the deep kernel machine? One can argue that an advantage of them over DNNs is its theoretical solidness. However, DNN also has theoretical support in the convergence and generalization, such as NTK.

---

> ### Author Response · Authors · 2024-11-23
>
> Thank you for the additional questions.
>
> Regarding the deep methods with kernels, they are also suited to the case where the number of samples are limited. An advantage of the deep methods with kernels is their connection with benign overfitting. As discussed in Section 6.2 in [3], we can learn models so that they overfit benignly if the number $L$ of layers is $L\ge 2$.
> Regarding the comparison with the theory of DNNs, not only the theoretical analysis of kernel methods is simpler, but also the deep model with kernels has an advantage on generalization bound.
> As discussed in Section 6.4 in [3], for the deep model with kernels, we can obtain a generalization bound whose dependency on the widths of the layers is smaller than those for DNNs.
> Different from obtaining the deep structure by the composition of kernels, such as [1] and [3], the deep structure in our case, discussed in Section 6, is obtained by the product of the proposed kernels, and is for improving the performance.
> We can extract both local and global dependencies at the same time by considering the product of the kernels.
> We can also apply the proposed kernels to the deep structure discussed in [1,3], and then we can have the same advantages of the deep methods discussed in [1,3], as discussed above.
>
> [3] Yuka Hashimoto, Masahiro Ikeda, and Hachem Kadri, "Deep learning with kernels through RKHM and the Perron-Frobenius operator", NeurIPS 2023.

---

> > ### Comment · Reviewer_jZ5N · 2024-11-24
> > **Comment**
> >
> > Thanks for your response. It deepened my understanding of the motivation for this work.

---

> > > ### Author Response · Authors · 2024-11-24
> > >
> > > Thank you very much for your comments and questions, and thank you for updating the score. If you have any additional comments, questions, and concerns, please let us know.

---

### Official Review · Reviewer_XMUi · 2024-11-03

**Soundness:** 2
**Presentation:** 3
**Contribution:** 2
**Rating:** 6
**Confidence:** 3

**Summary:**

This paper proposes a new class of positive definite kernels based on the spectral truncation. Detailed properties and examples have been discussed, and numerical results on both synthetic data and the MNIST dataset are presented.

**Strengths:**

The authors have introduced the basic properties of the proposed kernels and also investigated the generalization error. The presentation is clear.

**Weaknesses:**

Though the theoretical results are promising, two main questions remain unaddressed:
1. How can practical learning designs benefit from the new algebraic structures?
2. How can the development facilitate the kernel choices in practice? Section 6 seems to have some discussions on deep models, but a general development shall be data/task-dependent.

**Questions:**

See Weakness.

---

> ### Author Response · Authors · 2024-11-20
>
> Thank you very much for your constructive comments. We addressed you comments and questions and revised the paper. We summarize the answers below. Please see the global comment at the top of this page for more details of the motivation of this paper.
>
> **[W1] Benefit from the new algebraic structures**
> An important benefit of applying the noncommutative algebraic structure is that we can **go beyond separable and commutative kernels with low computational cost**.
> By virtue of the noncommutative structure, we can generalize commutative kernels, which enables us to induce interactions along data function domain.
> Since the kernel is function-valued, we can apply the framework of RKHMs and can obtain the final solution with the calculation of functions, which results in lower computational cost than the vvRKHS method with existing operator-valued kernels.
>
> **[W2] Kernel choices in practice**
> Applying the proposed kernels, we can extract both local and global dependencies of the output
> function on the input function.
> The proposed kernel has a parameter $n$, which control how much we focus on local dependencies.
> The optimal choice of $n$ depends on data and tasks.
> **If we want to focus on the global dependencies, then $n$ should be set as a small number.
> On the other hand, if we want to focus on local dependencies, then $n$ should be set as a large number.**

---

> > ### Comment · Reviewer_XMUi · 2024-11-21
> > **Thank you for your reply**
> >
> > I would like to clarify the questions: The kernel-related approaches are generally sensitive to kernel choices. On the one hand, it is inefficient to try many possible kernels. However, if the kernel choice depends on only a small subset of parameters (like in RBF kernels), then the family of RKHS is also restricted by such restricted choices. How can the characterization (potentially) address the kernel choices for practical data and problems?

---

> > > ### Author Response · Authors · 2024-11-22
> > >
> > > Thank you very much for the clarification of the question.
> > >
> > > We would like to emphasize that the proposed kernels are general and flexible. Especially, we have the following two points:
> > >
> > > - The proposed kernels are composed of scalar-valued kernels $\tilde{k}\_{i,j}$ and $\tilde{k}$ in Definition 3.2. We can choose any kernel for $\tilde{k}\_{i,j}$ and $\tilde{k}$, and the properties of the proposed kernels depend on that choice. If we choose a kernel with a small number of parameters for $\tilde{k}\_{i,j}$ and $\tilde{k}$, then the choice of  kernels is restricted, but if we choose a kernel with a large number of parameters (such as a weighted sum of multiple kernels), then by optimizing the parameters, we can obtain a better kernel for given data or tasks.
> > > - For function-valued kernels, in the same manner as the scalar-valued kernels, the weighted sum of positive definite kernels and product of positive definite kernels are also a positive definite kernel. Thus, we can consider multiple proposed kernels and combine them with weight parameters. In that case, we can also optimize the parameters to obtain a better kernel for given data or tasks.

---

> > > > ### Author Response · Authors · 2024-11-27
> > > >
> > > > Dear Reviewer XMUi,
> > > >
> > > > We updated the paper by adding the above points regarding the choice of kernels in Remark B.2 (colored in blue). We would appreciate if you could let us know whether the above answer addressed your question or not. Thank you for your time.

---

> > > > > ### Comment · Reviewer_XMUi · 2024-11-28
> > > > >
> > > > > The current manuscript has addressed my concerns. I have updated my score to reflect the improvement.

---

> > > > > > ### Author Response · Authors · 2024-11-28
> > > > > >
> > > > > > Thank you very much for your time to check the new version, and thank you for improving the score. If you have any additional comments, questions, and concerns, please let us know.

---

### Official Review · Reviewer_33dL · 2024-11-04

**Soundness:** 3
**Presentation:** 2
**Contribution:** 2
**Rating:** 3
**Confidence:** 3

**Summary:**

This paper proposes a new class of C*-algebra-valued positive definite kernels called spectral truncation kernels for vvRKHS. The noncommutativity, controlled by a truncation parameter n, allows for capturing interactions along the data function domain. The paper argues this enables a balance between representation power and model complexity, potentially leading to improved performance. A generalization bound is derived, highlighting the role of n in this tradeoff.

**Strengths:**

The paper introduces an approach to kernel design by leveraging the mathematical framework of C*-algebras and RKHM, offering a potentially powerful way to model complex data relationships. The theoretical analysis of generalization bounds provides valuable insights into the trade-off between representation power and model complexity, guided by the kernel's truncation parameter.

**Weaknesses:**

1. While the authors claim a computational advantage over vector-valued RKHSs (vvRKHSs) due to the linear dependency on output dimension m compared to cubic dependency in vvRKHSs, this advantage is not clearly demonstrated. The computational cost analysis lacks a direct comparison with vvRKHSs employing appropriate approximation techniques. For instance, the use of Nyström methods or random Fourier features could significantly reduce the computational burden of vvRKHSs, potentially negating the claimed advantage of spectral truncation kernels.

2. The deep model extension, while promising, lacks theoretical grounding. The analysis of representation power growth is based on a very specific construction and doesn't provide general insights into the behavior of deep networks with spectral truncation kernels.

3. The experimental results, while suggestive, are not compelling enough to validate the claimed advantages. The experiments are limited to synthetic data and a simplified MNIST task. More complex, real-world datasets with function-valued outputs are needed to assess the practical performance and demonstrate a clear advantage over existing methods.

**Questions:**

See weaknesses

---

> ### Author Response · Authors · 2024-11-20
>
> Thank you very much for your constructive comments. We addressed you comments and questions and revised the paper. We summarize the answers below.
>
> **[W1] Application of approximation techniques**
> Although we obtain a Gram matrix $G$ with function-valued elements for the proposed kernel, we can obtain the $\mathbb{C}^{N\times N}$ Gram matrix by evaluating it at a certain point $z$.
> Then, we can apply low-rank approximations such as the Nystrom method to $G(z)\in\mathbb{C}^{N\times N}$.
> In this way, we can reduce the computational cost with respect to $N$ for the proposed kernel, too.
> In the last paragraph of Section 5, we tried to insist that we can apply the Nystrom method to the proposed kernel, but we are sorry that the sentences were not clear.
> We revised the last paragraph of Section 5 to clarify the above point.
>
> The main goal of this paper is to propose a new class of function-valued kernels based on spectral truncation and investigate their fundamental properties.
> Thus, although we can apply some approximation methods to the proposed kernels, we focused more on the basic method without the approximation.
> In addition, we may develop approximation methods by taking advantages of the fact that the output of the kernels are functions.
> However, more detaild observation about the approximation methods that is specific to the proposed kernel is future work.
>
> **[W2] Analysis of the deep model**
> The goal of Proposietion 6.1 is not providing general analysis for deep models, but providing a certain archtecture that is effective in the sense of the deep model.
> For the deep model, we can construct the model by parameterizing the learnable values, and specifying the model architecture is important.
> The result in Proposition 6.1 provides a certain architecture of the deep model that achieves the exponential growth of the representation power with respect to the number of layers.
> Thus, this result is useful in constructing a model with a theoretical guarantee.
>
> **[W3] Additional experiments with more complex, real-world datasets**
> The motivation of this paper is to propose kernels going beyond the separable and commutative kernels with low computational cost and to theoretically investigate their fundamental properties.
> This work is the first attempt to achieve the above goal, and the goal of the numerical experiments in this paper is to confirm these fundamental properties, and more detailed experimental investigation is beyond the scope of this paper.
> Please see the global comment at the top of this page for more details of the motivation of this paper.

---

> ### Author Response · Authors · 2024-11-27
>
> Dear Reviewer 33dL,
>
> Thank you again for your comments. Regarding [W3], our main goal is to propose kernels going beyond the separable and commutative kernels with low computational cost and to theoretically investigate their fundamental properties.
> However, we agree that more experimental results are needed for showing the availability of the proposed kernels to pratical applications. For this purpose, we conducted an additional experiment.
>
> One potential application of the proposed kernel is the application to operator learning.
> In the framework of operator learning, we obtain a solution of a partial differential equation as an output from an input function (such as initial condition or parameter of the equation). Thus, we construct a model where both of the input and output are functions. Applying kernel methods to operator learning has been proposed [1]. We can construct the model by solving a kernel ridge regression task. Please see Appendix D for more details.
>
> We applied the proposed kernel to operator learning and obtained a higher performance than the existing method proposed in [1]. Please see Appendix D.1 (colored in blue) for more details.
>
> Although experiments for more complex cases are future work, we believe this result adresses your point [W3], and shows the possibility of the proposed kernels for futher applications.
> If you have any additional comments, questions, and concerns, please let us know.
>
> [1] Pau Batlle, Matthieu Darcy, Bamdad Hosseini, and Houman Owhadi, "Kernel methods are competitive for operator learning", Journal of Computational Physics, 496(1):112549, 2024.

---

> ### Author Response · Authors · 2024-12-02
>
> Dear Reviewer 33dL,
>
> Apologies for our continuous messages. Since the end of the discussion period is approaching, we would appreciate if you could let us know whether the above answer and additional experiment address your comments or not and let us know if you have additional questions, comments, and concerns related to these points. Thank you for your time.

---

> > ### Comment · Reviewer_33dL · 2024-12-02
> >
> > I thank the authors for their response. My primary concern with this paper lies in the unconvincing nature of its applications. They give two nice but simple demonstrations of their algorithm: data regression and an MNIST image reconstruction. In the context of operator learning (mentioned in the Appendix), the authors present an additional experiment with the Burgers' equation. While this shows some applicability, it remains a relatively simple example. If the proposed techniques were more novel, I would say that it is ok that the applications are underdeveloped. But since the techniques fairly build on well-established vector-valued RKHSs, I think the lack of developing applications for them weakens the paper's strengths.

---

> ### Author Response · Authors · 2024-12-02
>
> Thank you for your response.
>
> We would like to emphasize that our proposed technique is not on the framework of vvRKHSs, but on the framework of RKHMs [1]. As we explained in the global comment, the proposed kernels fill a gap between existing operator-valued kernels: separable and commutative kernels. For the cases $n=1$ and $n=\infty$, the proposed kernels are equivalent to the existing operator-valued kernels, where $n$ is the truncation parameter. However, if $1<n<\infty$,  they are different from existing operator-valued kernels, but they are **function-valued** kernels ($
> C^*$-algebra-valued kernels). We combined the proposed kernels with the framework of RKHMs. The reason why we can reduce the computational cost compared to the existing operator-valued kernels is that we proposed function-valued kernels and applied the framework of RKHMs. Please see table 1 and Section 5 for more details.
>
> The framework of RKHMs is a newly developed framework, and this paper gives a potential power of the application of RKHMs. This is the first paper that proposed function-valued kernels in the framework of RKHMs and showed its computational efficiency over the framework of vvRKHSs.
>
> [1] Yuka Hasimoto, Masahiro Ikeda, and Hachem Kadri, $C^*$-Algebraic Machine Learning − Moving in a New Direction, ICML 2024.

---

### Official Review · Reviewer_vkGt · 2024-11-06

**Soundness:** 3
**Presentation:** 3
**Contribution:** 3
**Rating:** 6
**Confidence:** 4

**Summary:**

This paper explores the recent subfield of positive definite kernels with values in a C*-algebra and RKHM (the correponding "RKHS" theory) . The whole work is motivated by going beyond the separable kernel widely used in vector-valued RKHS based on operator-valued kernels and benefit from with a better compute time when applying kernel “ridge” regression. The main interest of working with a C*-algebra is that it comes with a norm, a product and an involution, unifying operators and functions. In particular, the paper focuses on the C*-algebra of continuous functions and the case where inputs as well are elements of this C* algebra.  The paper is illustrated with the example of continuous functions on the 1D torus.  The authors propose a novel function-valued kernel, spectral truncation kernel, relying on the approximation of the multiplication operator with respect to x (defined in L2(T)) by leveraging a truncated spectral decomposition. The dimension of the truncated basis encodes a trade-off between the representation power and the model complexity. The resulting kernel also benefits from the noncommutativity of the approximated product and can be shown to converge. Applied on Kernel Ridge Regression in RKHM, this new kernel leads to a reduction of the complexity in time. It also comes with a generalization bound which is a direct instantiation of the result proven by Hashimoto et al. (2023). A deep architecture based on product (and not composition) of different kernel-based functions in RKHM is also presented. Experimental results study the behaviour of the approach with respect to the truncation parameter on  a toy dataset. An additional result on an inpainted image recovery problem built on MNIST data is also briefly presented.

**Strengths:**

This paper is following an original path in the kernel theory, exploring new schemes of kernels with values in a C* algebra. The work is certainly promising and of great interest for the kernel community. based on solid mathematical work, it opens a new way to tackle vector-valued or function-valued regression.
- Of special interest on the case of input and outputs that can be considered as functional (or vectors that can be seen as values of functions like images), the spectral truncation kernel allows for a drastically reduced computation cost in Kernel Ridge Regression while offering a great expressivity.
- It can be declined with various choices of ground kernels and its positive definiteness is studied
- Products of (function-valued)-functions based on those kernels provide a deep architecture.

**Weaknesses:**

Even though very interesting the paper suffers from different flaws: some concern the presentation and can be considered as relatively minor, while the others are more fundamental.
Weaknesses in the content:
*The motivation of the paper remains unclear: if the authors wish to go beyond the separable kernel in the general case of vector-valued functions, they should briefly discuss the limitations (which indeed exist) of the different operator-valued kernels such as the transformable kernels, the separable kernels or combination of them.  If the motivation is to use RKHM theory in the case of function-valued functions then it is of paramount importance to highlight what cannot be done with operator-valued kernels devoted to functions with outputs in Hilbert spaces of functions.
*Once a family of kernels is defined, in machine learning, we are interested on the ability of the machine learning algorithms to indeed take benefit from this kernel and provide a good solution to the ML task. So what is missing in this paper, is a discussion and an empirical study to determine when using those kernels are interesting compared to previous methods: does the complexity of the model make the algorithm more greedy in training data. I do not think that the actual generalization error bound really help to tell us that in precise terms.
My advice is thus to complete the paper with comparison with other (operator-valued) kernels and vv-RKHS. this has to be done  in the case of the current toy dataset but also in known functional regression data sets.
* Applicability and relevance of the methodology: a central question that is still not enough answered at the end of the paper is the following: on which family of problems, these spectral truncation kernels are relevant ?
For instance,  the use of function-valued kernels for inputs and outputs which are vectors should be discussed. I think it is important here to clarify this: images are by definition a discretisation of continuous maps (intensity of pixel in finite resomution) and then they can be seen as a set of values of a function taken on different observation points. There is a great interest at considering the functions as continuous functions.
 I do not think it is always meaningful for a vector to be encoded as a function of its coordinate index: can you comment on that ?
* Finally I do think that the paper would have sufficient content if it was restricted to function-valued functions. However, if images are tken as examples, then more convincing and complete comparison on image completion should be givne with more involved problems than weakly inpainted images. For the results given, do not say vv-RKHS comparions in the table say clearly the name of the matrix-valued kernel you used and try other kernels including more general operator-valued kernels for function-valued functions.

Weaknesses in the presentation of the paper.
The paper is in general not self content,  too much straightforward in its statements and very not enough precise in the presentation. It seems to me that an important work of re-writing is necessary, even though it is quite obvious that some efforst ahve been made here.
To give a few suggestions:
- rewrite the introduction with clear motivations and do not enter into partial details that cannot be understood at this stage (n < infty, n infty..)

- line 154 we jump into a comment about C(T) but previsously functions on the real torus were just an example. Now X = A ?
please say it !Moreover we cannot understand the sentence "however, by approximating ... by a Toeplitz matrix..;", we do not know yet that Toeplitz matrix will be involved here


- before line 174, say a word about the works of Van Suijlekom and explain the role of the Fejer function. The reader has to consult this apper to undertstand the construction. It is important in what follows when talking about convergence.

- in general, do not give proposition under the form of the sole formula but write a sentence introducing the property and a comment on what the proposition brings.
- generalization bound : clearly state as in the appendix that this result comes directly from previous literature (Maurer, 2016... Hasimoto et al. 2023)
-  It is crucial to introduce m when describing the observations at line 364.
- experiments :
what do you want to bring in terms of emprirical evidence: please present the experiments as an answer to the questions/motivation of the beginning of the paper

* after rebuttal: the paper has gained significant improvements and I am  increasing my score consequently.

**Questions:**

Overall I would be happy to increase the score of the paper (around 4) if answers are brought during the rebuttal.

- Please clarify the motivation and express the pros and cons with previous competing methods
- Give at least one example to make the reader understand the issue with commutativity and the benefit of non commutativity
- Identify as much as possible the family of problems that could in principle be tackled by this method
- Complete the toy experiments with a comparison with other function-valued regression methods
for the existing experiments, are the same curves observed when dealing with a very large data regime ?

Nearly all my questions have been answered in a satisfying way.

---

> ### Author Response · Authors · 2024-11-20
>
> Thank you very much for your constructive comments. We addressed you comments and questions and revised the paper. We summarize the answers below.
>
> **[Q1] Motivation, pros, and cons of the proposed method**
> The motivation of this paper is to propose kernels **going beyond the separable and commutative** (defined only with the pointwise calculation of functions or vectors) **kernels with low computational cost**.
> In the framework of vvRKHSs, transformable kernels and combinations of them with separable kernels have been proposed to go beyond separable and commutative kernels.
> However, an important shortcoming of them is the significant computational cost.
> To resolve this issue, we propose a function-valued kernel combined with the framework of RKHMs.
> Separable and commutative kernels are two extreme cases regarding the dependencies between input and output variables.
> Separable kernels identify dependencies between input and output variables separately, and cannot reflect information of input variables properly to output variables. **For the separable kernels, the output is determined only by the global information of the input**.
> On the other hand, **commutative kernels only identify the pointwise (completely local) dependencies**.
> **The proposed kernel fill a gap between separable and commutative kernels with lower computational cost** than existing operator-valued kernels.
> Indeed, we show that the proposed kernels converge to the commutative kernels as the truncation parameter $n$ goes to infinity.
> On the other hand, if $n=1$, then the proposed kernels are separable kernels.
> **If $n$ is small, the proposed kernel focuses more on global information of input functions.
> If $n$ is large, the proposed kernel focuses more on local information of input functions.**
> This feature is achieved by introducing the **noncommutativity** to typical commutative kernels by applying the theory of spectral truncation, and is described by the property of the Fejer kernel.
> The Fejer kernel goes to the delta function as $n$ goes to infinity.
> Please see Figure 2 in appendix B for more details about the Fejer kernel.
>
> As we discussed in Section 8, one limitation of the proposed kernels is that if we generalize them to those for more general functions than functions on the torus, then the theoretical analysis becomes more complicated, and the results in this paper may not valid for the generalized kernels.
> Although we can formally generalize the kernels, thorough theoretical analysis for the generalized kernels is future work.
>
> We revised the introduction and Section 3 to emphasize more on this motivation.
> In addition, we added Sections A and B in the appendix to explain it in more detail.
>
> **[Q2] Issue with commutativity and the benefit of noncommutativity**
> Commutative kernels only identify completely local relationship between the input function and the output function.
> The value of the output function at $z$ is determined only with the value of the input function at $z$.
> For example, if we have a time-series input $[x_1,\ldots,x_d]\in\mathcal{A}^d$ as explanatory variables and try to obtain an output function as a response variable, the values of $x_1(z),\ldots,x_d(z)$ at time $z$ is strongly related to the value of the output at time $z$, but may also related to $y(z+t)$ for $t\in [-T,T]$ for a small number $T$.
> In this case, the commutative kernels are not suitable for extracting the relationship between $x_1(z),\ldots,x_d(z)$ and the values of the output around $z$, not only at $z$.
> We documented detailed explanations of commutative kernels in Appendix A.
>
> Applying the theory of spectral truncation, we can induce the noncommutativity from commutative kernels and extract the relationship between $x_1(z),\ldots,x_d(z)$ and the values of the output around $z$.
> Moreover, we can control how much we will focus on local information by changing the value of $n$.
> If $n$ is large, then we can focus more on local information.

---

> > ### Author Response · Authors · 2024-11-20
> >
> > **[Q3] Problems that could in principle be tackled by the proposed method**
> > As you pointed out, considering the case where inputs and outputs are functions is important for the proposed kernel.
> > The application to image data is just an example, and there are more applications, which involves functions.
> > Although this work is the first attempt to applying the noncommutativity to go beyond separable and commutative kernels with low computational cost, it opens up these applications.
> > We list two examples below.
> >
> > 1. **Time-series data analysis**
> > We can regard a time-series as a function on a time space.
> > In many cases, a state at a certain time $z$ is influenced strongly by another state at the same time $z$, but also by the state around the time $z$.
> > Since commutative kernels focus only on local information, we cannot describe these two states with commutative kernels.
> > On the other hand, since separable kernels focus only on global information, we cannot describe the relationship of these two states at each time $z$.
> > By applying the proposed kernel, we can extract global information, but also can focus on local information.
> >
> > 2. **Operator learning**
> > In the framework of operator learning, we obtain a solution of a partial differential equation as an output from an input function (such as initial condition or parameter of the equation).
> > Thus, we construct a model where both of the input and output are functions.
> > Applying kernel methods to operator learning has been proposed [1].
> > We can construct the model by solving a kernel ridge regression task.
> > Another well-known operator learning method is neural operator.
> > In the framework of neural operators, we apply integral operators to extract global information and apply local linear operators and local activation functions to extract local information.
> > The proposed kernel enables us to do similar procedures for the operator learning with kernels.
> > By considering the product of multiple proposed kernels with different values of $n$ or deep model with the proposed kernels with different values of $n$, we can extract both global and local information; we can extract global information using the kernel with small $n$ and extract local information using the one with large $n$ in the model.
> >
> > We mentioned about the application in Section 8 and added Appendix D for explaining above examples.
> >
> > [1] Pau Batlle, Matthieu Darcy, Bamdad Hosseini, and Houman Owhadi, "Kernel methods are competitive for operator learning", Journal of Computational Physics, 496(1):112549, 2024.
> >
> > **[Q4] Comparison with other function-valued regression methods with very large data**
> > The motivation of this paper is to propose kernels going beyond the separable and commutative kernels with low computational cost and to theoretically investigate their fundamental properties.
> > The goal of the numerical experiments in this paper is to confirm these fundamental properties, and more detailed experimental investigation is beyond the scope of this paper.
> > In addition, an advantage of kernel methods is that they are valid even with a limited number of samples.
> > Thus, we focused on the case of small numbers of samples in the experiments.
> >
> > **[W1] The matrix-valued kernel used in the experiment**
> > In the experiment in Subsection 7.1, we compared the proposed kernel with an existing typical nonseparable kernel (combination of a transformable kernel and separable kernel) proposed by Lim et al. (2015).
> > As we stated in Appendix A, the transformable kernels are described using an integral operator, and the matrix valued kernel used in Subsection 7.1 is regarded as a proper discretization of the operator-valued kernel.
> > We think it is natural to use this operator-valued kernel to functional data, and it is suitable for the comparison with the proposed kernels.
> > We can see that the proposed kernels outperform the above operator-valued kernel and the computational cost for the proposed kernel is also lower than the operator-valued kernel, which shows the advantages of the proposed kernel over typical nonseparable and noncommutative operator-valued kernels.

---

> > > ### Author Response · Authors · 2024-11-20
> > >
> > > **Regarding the comments on the presentation**
> > > Thank you for your suggestion about the presentation. We also revised the paper based on your comments on the presentation.
> > >
> > > **[P1] Rewriting the introduction with clear motivations**
> > > We rewrote the introduction so that the motivation of the proposed kernel (beyond separable and commutative kernels with lower computational cost than existing operator-valued kernel with the framework of vvRKHSs) becomes clear.
> > > We also added a table (Table 1) to explain the difference between the proposed kernel and existing kernels.
> > >
> > > **[P2] Clarification of the data space $\mathcal{X}$ and about the sentence "however, by approximating ... by a Toeplitz matrix..;"**
> > > Section 2.2 is purely for spectral truncation, and is not directly related to the kernel methods.
> > > Thus, we added the explanation of $\mathcal{X}$ just before Example 3.1.
> > > We also deleted the word "Toeplitz" in the second sentence in Section 2.2.
> > >
> > > **[P3] The role of the Fejer kernel**
> > > We added the explanation of Fejer kernel before Proposition 2.6.
> > >
> > > **[P4] About the presentation of Lemma 3.9**
> > > We added some words in the statement of Lemma 3.9.
> > >
> > > **[P5] References for deriving the generalization bound**
> > > We do not agree that this result comes directly from previous literature.
> > > We confirmed that the results about generalization bound is also valid for the $C^*$-algebra-valued (function-valued) regression problem.
> > > Also, we showed that the operator-norm of the Toeplitz matrix $R_n(x)$ grows as $n$ becomes large.
> > > We agree that we need to refer previous literature for deriving the generalization bound in the main text, too.
> > > Thus, we added the references before Theorem 4.1.
> > >
> > > **[P6] Introducing m in Section 5**
> > > We added an explanation about $m$ at the beginning of Section 5.
> > >
> > > **[P7] Presenting the experiments as an answer to the questions/motivation of the beginning of the paper**
> > > We added the goals of the experiments in Section 7.

---

> > > > ### Comment · Reviewer_vkGt · 2024-11-25
> > > > **Feedback on the rebuttal**
> > > >
> > > > I appreciate the answers provided by the authors and acknowledge the improvements proposed in the paper.
> > > > In fact the presentation of the motivation, the new insights at different places of the paper, and the new discussion and clearer references significantly enhances the quality of the paper, yiedling me to improve my score.

---

> > > > > ### Author Response · Authors · 2024-11-25
> > > > >
> > > > > Thank you very much for your constructive comments and questions in your review, and thank you for improving the score. Your comments helped us to improve our paper. If you have any additional comments, questions, and concerns, please let us know.

---

> ### Author Response · Authors · 2024-11-27
>
> Dear Reviewer vkGt,
>
> We just want to let you know that since the deadline of revising the paper is approaching, we updated our paper by adding additional experimental results about operator learning in Appendix D.1 (colored in blue). We think this modification is related to your comments about the practical applications. Thank you again for your time and valuable comments.

---

### Author Response · Authors · 2024-11-20

**To all the reviewers**
We thank all the reviewers for their constructive comments and questions. We addressed them and revised the paper based on the comments. The revised parts are colored in red. The answer of each comment and question is posted as a comment for each reviewer.

We would like to emphasize that this work is the first attempt to incorporate the noncommutativity of the product combined with spectral truncation to go beyond typical kernel choices.
The theoretical results open up a new direction of the applications of the proposed noncommutative kernels.
Here, we would like to summarize the advantages of the proposed kernels below:

1. **Control of local and global dependencies**
For kernel methods with function- or vector-valued outputs, one big challenge is the choice of kernels.
In the existing framework of vvRKHSs, we need operator-valued kernels instead of scalar-valued kernels.
Typical choices are separable kernels and commutative kernels.
Although they are computationally efficient, they are two extreme cases in the sense of dependencies of input functions or vectors on output functions or vectors.
**Separable kernels focus only on global dependencies and commutative kernels focus only on local dependencies.
The proposed kernels fill a gap between these two kernels by virtue of the noncommutativity.**
The noncommutativity is parameterized by a natural number $n$, and we showed that this parameter controls how much amount of local dependencies are focused on.

2. **Computational efficiency**
To fill a gap between separable and commutative kernels, other operator-valued kernels in the framework of vvRKHSs have been proposed.
The proposed kernel combined with the framework of RKHMs are **computationally more efficient than these existing kernels with vvRKHSs**.

3. **Control of the representation power and the model complexity**
The parameter $n$ in the proposed kernel also controls the representation power and the model complexity, which leads the performance enhancement.

For more details, please see the answers of the reviewers and Appendices A, B, and D of the revised version of the paper.

---

### Meta-Review · Area_Chair_hadh · 2024-12-15

**Metareview:**

This paper utilizes the tool of C-algebra to develop new methods of kernel machines, receiving certain recognition from the reviewers. However, the current paper suffers from obvious weaknesses that prevent me from recommending acceptance. To me specific, the experimental setup, consisting of a synthetic dataset and a simple dataset called MINIST, is too weak to be convincing.

**Additional Comments On Reviewer Discussion:**

After author-reviewer discussions, the major concern remains unsolved:

Reviewer vkGt, who strongly suggests rejection, states that “I thank the authors for their response. My primary concern with this paper lies in the unconvincing nature of its applications. They give two nice but simple demonstrations of their algorithm: data regression and an MNIST image reconstruction. In the context of operator learning (mentioned in the Appendix), the authors present an additional experiment with the Burgers' equation. While this shows some applicability, it remains a relatively simple example. If the proposed techniques were more novel, I would say that it is ok that the applications are underdeveloped. But since the techniques fairly build on well-established vector-valued RKHSs, I think the lack of developing applications for them weakens the paper's strengths”.

The AC has indeed checked the paper and agrees that the reviewer is spot on.

---

### Decision · Program_Chairs · 2025-01-22

Reject